# The Roles of Epigenetic Regulation and the Tumor Microenvironment in the Mechanism of Resistance to Systemic Therapy in Hepatocellular Carcinoma

**DOI:** 10.3390/ijms24032805

**Published:** 2023-02-01

**Authors:** Kyoko Oura, Asahiro Morishita, Sae Hamaya, Koji Fujita, Tsutomu Masaki

**Affiliations:** Department of Gastroenterology and Neurology, Faculty of Medicine, Kagawa University, Kita 761-0793, Kagawa, Japan

**Keywords:** hepatocellular carcinoma, drug resistance, molecular target agent, tyrosine kinase, sorafenib, regorafenib, lenvatinib, systemic therapy, immune checkpoint inhibitor, tumor microenvironment

## Abstract

Primary liver cancer is the sixth most common cancer and the third most common cause of cancer-related deaths worldwide. Hepatocellular carcinoma (HCC) is a major histologic type with a poor prognosis owing to the difficulty in early detection, the chemotherapy resistance, and the high recurrence rate of the disease. Despite recent advancements in HCC prevention and diagnosis, over 50% of patients are diagnosed at Barcelona Clinic Liver Cancer Stage B or C. Systemic therapies are recommended for unresectable HCC (uHCC) with major vascular invasion, extrahepatic metastases, or intrahepatic lesions that have a limited response to transcatheter arterial chemoembolization, but the treatment outcome tends to be unsatisfactory due to acquired drug resistance. Elucidation of the mechanisms underlying the resistance to systemic therapies and the appropriate response strategies to solve this issue will contribute to improved outcomes in the multidisciplinary treatment of uHCC. In this review, we summarize recent findings on the mechanisms of resistance to drugs such as sorafenib, regorafenib, and lenvatinib in molecularly targeted therapy, with a focus on epigenetic regulation and the tumor microenvironment and outline the approaches to improve the therapeutic outcome for patients with advanced HCC.

## 1. Introduction

### 1.1. Hepatocellular Carcinoma (HCC)

Primary liver cancer is the sixth most common cancer and the third most common cause of cancer-related deaths worldwide [1,2]. The global incidence of primary liver cancer continues to soar, and it is estimated that more than one million patients will be diagnosed with liver cancer annually by 2025 [3]. HCC is the most common histological type, accounting for approximately 90% of all primary liver cancers, and is one of the cancer types with a poor prognosis, owing to the difficulty in early detection, resistance to chemotherapy, and a high recurrence rate [4]. Hepatitis B virus (HBV) infection is the major risk factor for HCC, accounting for 60% of cases in Asia and Africa and 20% in Western countries [3]. Hepatitis C virus (HCV) infection is the most common cause of HCC in Western countries and Japan, but this trend has been declining as more patients achieve sustained biological response (SVR) with the widespread use of direct-acting antiviral therapy against HCV. However, patients with liver cirrhosis caused by HCV remain at risk of developing HCC by more than 2% per year after achieving SVR [5,6]; therefore, the incidence of HCC caused by HCV is expected to persist for some time. Furthermore, cases of non-alcoholic steatohepatitis (NASH) associated with certain metabolic syndromes, including diabetes mellitus, hypertension, dyslipidemia, and obesity, are rapidly increasing as an etiological factor of HCC, especially in Western countries.

Although there have been significant advances in the prevention and diagnosis of HCC in recent years, more than 50% of all HCC patients are still diagnosed with Barcelona Clinic Liver Cancer (BCLC) stage B or C [3]. The 5-year survival rate for early stage HCC exceeds 70%, whereas the median survival time for patients with advanced HCC treated with systematic therapies is 1–1.5 years [7]. Patients with advanced HCC with major vascular invasion or extrahepatic metastases are classified as stage C according to the BCLC staging system, and systemic therapies are recommended. Even in BCLC stage B, systemic therapies have increasingly been recommended for patients with intrahepatic lesions that appear to have a limited response to conventional transcatheter arterial chemoembolization, such as those with HCC exceeding the up-to-7 criteria [8].

### 1.2. Systemic Therapy for Advanced HCC

Six chemicals have been approved as systemic therapies for unresectable HCC based on Phase III clinical trials, sorafenib [9], regorafenib [10], lenvatinib [11], cabozantinib [12], ramucirumab [13], and atezolizumab plus bevacizumab (atezo + bev) [14]. These are expected to expand the treatment options in the future. Molecular target agents (MTAs) are effective against various cancer types, including HCC, but, in many cases, even if initially effective, the drugs gradually become resistant and cause recurrence, which is a serious problem in clinical management. Important mechanisms underlying the acquisition of the resistance to MTAs in cancer are alterations in the target gene (gene amplification, gatekeeper mutations, and secondary mutations), activation of associated pathways, downstream activation of the target, epithelial–mesenchymal transition (EMT), and transformation of cancer stem cells. Elucidation of the mechanisms underlying MTA resistance in cancers and the appropriate responses to the issue of MTA resistance will contribute to improved outcomes in the multidisciplinary treatment of advanced HCC.

This review summarizes recent findings regarding the mechanisms of HCC resistance to MTAs, including sorafenib, regorafenib, and lenvatinib, and outlines the approaches suitable for improving the outcomes of systemic therapies in patients with advanced HCC.

## 2. Epigenetic Regulation of Drug Resistance in HCC

Epigenetic regulation involves gene regulatory mechanisms that are encoded by heritable modifications to genomic and chromatin components, but the underlying deoxyribonucleic acid (DNA) sequences are unchanged [15]. Epigenetic mechanisms regulate various biological processes, including cell growth and differentiation [16,17]. Epigenetic changes affect gene regulation by altering the DNA structure, and they have been observed in several cancers such as HCC [18]. Common epigenetic regulatory mechanisms include DNA methylation, chemical modification of histones, and regulation by non-coding ribonucleic acid (ncRNA) [19], which are associated with various processes in HCC, such as carcinogenesis, progression, metastasis, and angiogenesis (Figure 1). Epigenetic modifiers consist of writers, readers, and erasers. Epigenetic writers are a group of enzymes that add methyl or acetyl groups to histone proteins and include tyrosine kinases, serine-threonine kinases, DNA (cytosine-5)-methyltransferases (DNMT), and enzymes such as histone acetyltransferases (HAT) and histone lysine methyltransferases (HMT). Epigenetic readers are proteins that recognize functional modifications of epigenetic marks placed on DNA or histones with binding domains for covalent modifications, including bromodomains involved in histone acetylation, chromodomains involved in histone methylation, and methyl CpG-binding proteins, which allow modulation of chromodomain conformations through dynamic integrated signals [20]. Epigenetic erasers are a group of enzymes that eliminate epigenetic modifications from histone, these include histone deacetylases (HDACs) and histone demethylases. In HCC, DNA methylation and histone modification levels have been shown to be markedly elevated during the progression from liver cirrhosis to carcinogenesis [21]. These changes have also been observed in non-tumor tissues with chronic liver injury and cirrhosis, and can serve as a prognostic marker for the progression and recurrence of HCC [22]. A deeper understanding of HCC-associated epigenetic modifications could lay the foundation for elucidating the emergence of resistance to MTAs and the potential strategies to overcome it.

### 2.1. DNA Methylation

DNA methylation is one of the best characterized epigenetic modifications that plays a crucial role in regulating gene expression. The addition of a methyl group at the fifth carbon position of cytosine forms 5-methylcytosine (5mC). 5mC regulates many biological functions in the genome, but only 3–4% of all cytosines in the genome are methylated [23]. Methylation occurs only at the CpG sites, where a cytosine nucleotide occurs on the 5′ side of a guanine nucleotide. 5mC tends to be spontaneously deaminated, leading to a cytosine-to-thymine transition. CpG islands that are generally found in the promoters of genes are not methylated in normal cells, but methylated CpG islands are enriched in suppressed gene regions such as repetitive sequences, inactivated genes on the X chromosome, imprinted genes, and certain tissue-specific genes. DNA methylation is an epigenetically heritable signal that accompanies the enriched chromatin structure and maintains gene silencing. The epigenetic DNA methylation marker 5mC is mediated by enzymes such as DNMT1, DNMT3A, and DNMT3B [18]. DNMT1, a conversion enzyme, methylates the cytosine bases on the unmethylated strand of the hemimethylated DNA to preserve methylation patterns during DNA replication. DNMT3A and DNMT3B play important roles in the developmental processes by methylating unmethylated CpG dinucleotides. Methyl-binding domain protein (MBD) binds HDACs and chromatin remodeling enzymes, and mobilization of MBD leads to the silencing of DNA methylation. Epigenetic regulation of transcription involves cross-talk between DNA methylation, chromatin remodeling enzymes, and histone modifications. Dysfunction in DNA methylation has been shown to be closely associated with carcinogenesis, autoimmune diseases, and fibrosis diseases [24]. In particular, genome-wide hypomethylation and localized anti-methylation of tumor suppressor gene promoters are hallmarks of various cancers, including HCC [25], and methylation abnormalities are potential biomarkers and therapeutic targets in HCC.

DNA hypomethylation in HCC is caused by genomic instability [26], frequent mutations, and transitions occurring at inactive sites in chromatin regions [27], often in specific CpG islands, repeated DNA sequences, and intergenic regions. Such global methylation pattern in HCC has been shown to upregulate transcription factors, and previous studies have identified overexpressed transcription factors with enhancer methylation patterns that can be therapeutic targets in HCC. Genome sequencing analysis has shown that CCAAT/enhancer-binding protein-beta (C/EBPβ) enhancer, a transcription factor overexpressed in HCC patients, is repeatedly hypomethylated throughout the genome, and such hypomethylation has been shown to correlate with C/EBPβ overexpression and prognosis in HCC patients [28]. The enhancer of C/EBPβ reactivates enhancer RNA bound to its enhancer, forming a self-reinforcing enhancer-target loop, thus enabling the transcription of C/EBPβ. Furthermore, the loss of this enhancer has been shown to markedly suppress various oncogenes and hepatocarcinogenesis.

Other hypomethylated genes, such as pyruvate dehydrogenase kinase 1 (PDHK1) and phosphoglycerate kinase 1 (PGK1), are also known to be associated with HCC. A study examining PGK1 messenger RNA (mRNA) levels and DNA methylation in normal and 34 types of cancerous tissues has shown that PGK1 mRNA expression decreases following the hypomethylation of its promoter region, and this is associated with poor prognosis in several cancer types, including HCC [29]. In HCC, PDHK1 T338 and PGK1 S203 phosphorylation levels are positively correlated and are associated with shorter overall survival (OS). Controversially, DNA hypermethylation at CpG islands promotes the silencing of tumor suppressor genes, and many hypermethylated gene promoters, such as the promoter regions of adenomatous polyposis coli (APC) gene and cyclin-dependent kinase inhibitor 2A (CDKN2A) gene, have enabled researchers to distinguish tumor tissues from non-tumor liver tissues in patients with HCC [30]. In addition, another genome-wide methylation profiling study in HCC and normal liver tissues has shown that significant methylation differences are present at 13% loci between HCC and proximal normal tissues, indicating that the methylation at promoter CpG islands is enriched in HCC tissues [31]. An elevated expression of enzymes such as DMNT also causes changes in DNA methylation, and a recent study indicated that DMNT1 is relatively overexpressed in HCC cell lines and tissues, and is associated with poor prognosis [32]. Furthermore, combined inhibition of DNMT1 together with G9a HMT, which is associated with histone modification, synergistically inhibits the growth of HCC, suggesting that combined DNMT1/G9a targeting is a promising strategy in the treatment of HCC.

Further, there are several reports on DNMT-mediated epigenetic changes with regulation of HCC metastasis. With respect to the role of DNMT1, the knockdown of non-collagenous bone matrix protein osteopontin in CD133+/CD44+ HCC cells, which have cancer stem cell properties, suppresses DMNT1 expression and promotes HCC metastasis [33]. Epigenetic upregulation of c-Met, the receptor for hepatocyte growth factor (HGF) is associated with HCC progression and metastasis in the TME, and it was shown that a significant decrease in DNA methylation during the hematogenous metastasis of HCC correlated with increased c-MET expression in circulating tumor cells [34]. Other reports showed that the induction of DNMT1 expression by HGF led to the DNA hypermethylation of tumor suppressor genes such as myocardin, pannexin 2, and LIN homeobox 9 genes, which was associated with HCC metastasis [35]. DNMT3 has also been shown to promote HCC metastasis and invasion by epigenetic regulation of the metastasis-associated protein 1 (MTA1) gene, and in HBV-associated HCC, HBV X mobilizes DNMT3a and DNMT3b to increase promoter methylation and enhance MTA1 expression [36].

Recently, DNA methylation has also been reported to be important in the acquisition of drug resistance, and chemotherapy is known to cause epigenetic changes in gene expression without inducing genetic mutations. In a previous study that established a cell line resistant to morpholino anthracycline derivatives in human myeloid leukemia cell lines and analyzed the underlying drug resistance mechanism, methylation-specific restriction enzyme analysis revealed that the CpG of the topoisomerase IIα gene is abnormally methylated in the resistant cell line [37]. Furthermore, the genomes of the drug-resistant cell lines are globally methylated, and genes involved in immune responses and gene silencing have been identified as the source of methylation-related gene expression changes [38]. In HCC, changes in DNA methylation at the EMT gene promoters may be important in the acquisition of drug resistance. DNA methylation-driven EMT underlies resistance to first-line sorafenib treatment in patients with advanced HCC [39]. In patients whose disease was initially controlled but later developed drug resistance, consistent changes were observed in the EMT-driven DNA methylation patterns; however, no such changes were observed in the group that continued to respond well to sorafenib treatment. In addition, the evaluation of the DNA methylation status of the EMT gene promoters in liquid biopsy should be considered a biomarker for evaluating the response to sorafenib treatment. In an in vivo study using subclones of drug-resistant human HCC cells established by long-term treatment with vascular endothelial growth factor (VEGF) receptor inhibitors and serial transplantation into immunocompromised mice, thymosin beta 4 (Tβ4), a G-actin monomer binding protein, is enriched in resistant HCC cells, and DNA demethylation and histone H3 activation at its promoter region result in an aberrant expression of Tβ4, promoting the growth of sorafenib-resistant tumors [40]. In summary, it is important to consider the regulation of epigenomic alterations, particularly DNA methylation, as potential mechanisms underlying refractoriness or resistance in advanced HCC.

### 2.2. Histone Modifications

Histone proteins undergo diverse post-translational modifications, including acetylation, methylation, ubiquitination, and phosphorylation. Histone acetylation in particular enhances gene expression by relaxing chromatin structures and making DNA more accessible to transcription factors [41,42], and HATs and HDACs are the major enzyme families involved in this process. HATs acetylate certain histone tail lysine residues, and this modification is associated with transcriptional activation by facilitating the binding of transcription factors such as E2F and p53 to the chromatin, and by promoting the activity of RNA polymerase II. In contrast, HDACs function to eliminate the acetyl group, which consequently restores the positive charge of lysine residues in the histone tails, and thereby stabilizes the high degree of chromatin condensation, which suppresses transcription. Additionally, deacetylation promotes DNA–histone interactions and condenses the chromatin structure [43]. Transcriptional activators mobilize HATs and other chromatin-remodeling enzymes to the promoter region of a specific gene.

There is growing evidence for a role of histone modifications in several cancer types, and epigenetic regulation via histone modifications is known to contribute to carcinogenesis. A study using lymphoblastic leukemia cell lines resistant to nucleotide metabolic antagonists has shown that there is no difference in the methylation status of CpG alleles in the deoxycytidine kinase promoter between resistant and parental cell lines, but the total histone acetylation and the acetylation of histone H3 and H4 are significantly lower in the resistant cell lines than in the parental cell lines [42]. Alterations in HAT and HDAC activities have also been observed in several cancer types, and the gene encoding the HAT E1A binding protein p300 (EP300) is mutated in epithelial cancers, suggesting its function as a tumor suppressor gene. A study that sequenced DNA-matched tumors from patients with B-cell non-Hodgkin’s lymphoma showed that genes involved in histone modifications are frequently subject to somatic mutations [44]. Patients with diffuse large B-cell lymphoma have somatic mutations in myeloid/lymphoid 2 encoding an HMT and in myocyte enhancer factor 2 B, a calcium-regulated gene that cooperates with the cAMP-responsive element-binding protein and EP300 to acetylate histones, suggesting that epigenetic changes play an important role in hematologic tumorigenesis.

In histone modifications in HCC, research on histone H3 methylation and acetylation has been the most extensive. Abnormally high levels of H3K4 trimethylation and H3 acetylation were observed, and H3K27 trimethylation was low in promoters in the vasohibin 2 (VASH2) gene, which functions as a growth factor in HCC [45]. Interestingly, suppression of VASH2 expression inhibited HCC proliferation, which induces apoptosis. In another report on histone H3 acetylation it was shown that insulin induced major transcriptional factors such as sterol regulatory element-binding transcription factor 1c (SREBP-1c) and carbohydrate responsive element-binding protein (ChREBP) binding with sterol regulatory elements (SRE) or carbohydrate-responsive elements (ChORE) of the fatty acids synthase (FASN) promoter and induces FASN expression in normal tissue, while the hyperacetylation of histone H3 and H4 impaired SREBP-1c-SRE and ChREBP-ChORE binding on the FASN promoter and HCC became insulin resistant [46]. An in vitro experiment has shown that HCC cells have relatively lower nucleosome density with histone H3K9 acetylation than controls, regardless of transcriptional activation status, which may play an important role in initiating HCC development [47]. Alternatively, for H3K methylation, it has been shown that the higher the level of H3K4 trimethylation in HCC, the worse the prognosis for HCC [48]. The histone methyltransferase mixed-lineage leukemia (MLL) causes H3K4 trimethylation, and the MLL-E-twenty-26 transcription factor 2 complex occupies the matrix metallopeptidase 1 (MMP1)/MMP3 gene promoter, resulting in the activation of the MMP1/MMP3 expression. This means that MLL-mediated H3K4 trimethylation is required for HCC proliferation and metastasis through HGF [49]. In another report, HBV X protein was shown to promote hepatocarcinogenesis in a cellular model by inducing the deposition of H3K9me3 at the p16 promoter via the downregulation of the demethylase jumonji domain containing 2B genes, which promotes the repression of the p16 gene [50].

Acyl-CoA thioesterase 12 (ACOT12), expressed in the liver, is an essential enzyme for the hydrolysis of acetyl-CoA and regulates the transfer of acetyl groups from acetyl-CoA to lysine residues by HATs. A recent study has reported that ACOT12 expression is markedly decreased in HCC and is associated with metastasis and poor prognosis [51]. Further experiments using cell lines and xenograft models have also shown that ACOT12 suppresses EMT and inhibits HCC metastasis by regulating intracellular acetyl-CoA levels and histone acetylation. Another study with HCC cell lines showed that HAT1 is also involved in catalyzing succinylation at the lysine 122 of H3 in the presence of its cofactor, succinyl-CoA, which increases phosphoglycerate mutase 1 enzyme activity and promotes tumor progression. The upregulation of HAT involved in the acetylation of lysine 16, a histone modification involved in transcriptional activation, also induces microvascular invasion [52].

In contrast, HDACs are known to play an important role in determining various features of HCC, such as progression and drug resistance. HDACs are classified into either one of the eleven zinc-dependent HDACs (HDAC1–HDAC11) or zinc-independent HDACs, which include sirtuin 1 (SIRT1), SIRT2, and SIRT7 [53]. A meta-analysis of nine studies has found that increased expression of the histone demethylase SIRT1 correlates with a larger tumor size and higher p53 expression and is associated with poor OS and disease-free survival in HCC patients [54]. Increased expression of HDAC1 and HDAC2 in HCC tissues is associated with low expression of fructose-1,6-bisphosphatase (FBP1), which is involved in glucose metabolism and poor prognosis in HCC patients [55]. HDAC inhibition in HCC restores FBP1 expression, glucose depletion, and lactase secretion and halts tumor growth in vitro and in vivo.

The suppression of these molecules involved in histone modification is a potential strategy for HCC treatment and management. The HMT enhancer zeste homolog 2 (EZH2) represses gene transcription through histone 3 lysine 27 trimethylation (H3K27me2) and is overexpressed in HCC due to gene silencing via H3K27me2 [56]. Other previous reports have suggested that the overexpression of H3K27me3 in HCC is associated with increased tumor size, vascular invasion, and metastasis [57], and elevated H3K27me3 levels in HCC are associated with poor OS [48]. G9s is another HMT that is often upregulated in HCC. G9a induces the epigenetic silencing of the tumor suppressor retinoic acid receptor responder protein 3, and increased G9a expression is associated with HCC progression and poor pathological features, such as vascular invasion, tumor microsatellites, and absence of tumor encapsulation [58]. Functional analysis has shown that the inactivation of G9a markedly inhibits H3K9 demethylation and suppresses HCC growth and metastasis in vitro and in vivo. Another report has also demonstrated that G9a and DNMT1, alongside their molecular adaptor, ubiquitin-like with PHD and RING finger domains 1, are overexpressed in HCC and are associated with poor prognosis [32]. The inhibition of G9a and DNMT1 synergistically suppresses HCC growth. The lead compound that regulates this mechanism restores FBP1 expression, which is epigenetically repressed in HCC, and thus, could be a suitable target in the therapeutic strategy for HCC.

### 2.3. Non-Coding RNA

Non-coding RNAs (ncRNAs) are RNAs that do not encode proteins or peptides. They are classified into small ncRNAs of 20–30 bases and long ncRNAs (lncRNAs) of several hundred kilobases (kb). Small ncRNAs include microRNAs (miRNAs), small interfering RNAs, and PIWI-interacting RNAs. ncRNAs are involved in various cellular functions, such as proliferation, cell-cycle progression, and apoptosis [59]. miRNAs and lncRNAs in particular regulate gene expression through transcription, translation, and protein functions at various levels and have been reported to function not only as biomarkers but also as therapeutic targets and drug resistance [60,61,62,63,64,65].

lncRNAs are derived from intergenic ncRNAs, specific antisense genes, promoter regions, introns, and the 3′ untranslated region (3′-UTR) [66,67]. Abnormal expression of several lncRNAs has been observed in HCC, and these lncRNAs have been reported to interact with DNA, RNA, and proteins to form complexes that regulate the expression of target genes. These lncRNAs are associated with clinical characteristics of HCC, such as metastasis, prognosis, recurrence, resistance to treatment, recurrence, and prognosis, through various mechanisms [67,68,69,70]. Lung-cancer-associated transcript 1 (LUCAT1) is an lncRNA known to regulate growth and metastasis in various cancer types. LUCAT1 expression is also enhanced in the tissues and cells of HCC, and loss- and gain-of-function studies have shown that LUCAT1 promotes growth and metastasis in HCC [68]. lncRNA MCM3AP antisense RNA 1 (MCM3AP-AS1) was also overexpressed in HCC tissues and cell lines and positively correlated with large tumor size, high tumor grade, advanced tumor stage, and poor prognosis, indicating that the knockdown of MCM3AP-AS1 inhibits HCC growth. Furthermore, MCM3AP-AS1 directly binds to miR-194-5p and promotes the expression of its target gene forkhead box A1 (FOXA1), which was the anti-tumor mechanism of HCC [69]. Thus, many lncRNAs involved in HCC progression have been identified, and the use of several anti-tumor compounds that target these lncRNAs may be the best therapeutic strategy. Overexpression of lncRNA metastasis associated with lung adenocarcinoma transcript 1 (MALAT1) was observed in HCC, and the lncRNA promoted HCC proliferation via the overexpression of SIRT1 [71], it was also shown that gallic acid downregulates MALAT1, resulting in Wnt/β-catenin signal inhibition and the suppression of HCC progression [72]. In addition, there are several reports on the anti-tumor effects of targeting lncRNAs with melatonin, which is used in the treatment of HCC. Melatonin was shown to increase lncRNA RAD51 antisense RNA 1 expression, mediate drug sensitivity, and inhibit HCC progression [73]. Another study showed that melatonin promotes FOXA2 expression and upregulates lncRNA Carbamoyl-phosphate synthetase 1 during the downregulation of hypoxia-induced factor-1α (HIF-1α), inhibiting EMT and HCC carcinogenesis [74], which may be a therapeutic strategy to target IncRNA.

Several lncRNAs are associated with the mechanisms of resistance to cytotoxic drugs in HCC. An HCC-associated lncRNA, HANR, is overexpressed in HCC tissues and is associated with poor prognosis in patients [75]. Silencing lncRNA HANR has been linked to the inhibition of cell proliferation, induction of apoptosis, and increased sensitivity to doxorubicin in HCC [75]. The altered expression of HANR affects the sensitivity of HCC to doxorubicin by suppressing glycogen synthase kinase-3 beta (GSK3β) phosphorylation and upregulating GSK3β total protein expression. Conversely, the overexpression of the lncRNA activated in renal cell carcinoma with sunitinib resistance (lncARSR), which is markedly upregulated in HCC and is associated with disease progression, has been shown to activate the phosphatidylinositol 3-kinase (PI3K)/Akt pathway by competing with phosphatase and tensin homolog (PTEN) mRNA, promoting doxorubicin resistance in HCC [76]. Another report involving oxaliplatin resistance showed that lcnRNA nuclear receptor subfamily 2 group F member 1-antisense RNA 1 (NR2F1-AS1) is overexpressed in oxaliplatin-resistant HCC cells and tissues, contributing to drug resistance; NR2F1-AS induces the member of the ATP-binding cassette transporter superfamily associated with multidrug resistance via the endogenous sponge miR-363, and attenuates oxaliplatin sensitivity in HCC [77]. Further, the lncRNA termed highly upregulated in liver cancer (HULC) induces autophagy by downregulating SIRT1 expression and attenuating sensitivity to oxaliplatin and 5-fluorouracil (5-FU) in HCC [78]. The lncRNA metastasis-associated MALAT1 is also associated with 5-FU resistance. This lncRNA is overexpressed in 5-FU- and adriamycin-resistant HCC cells, indicating that MALAT1 knockdown could overcome 5-FU and adriamycin resistance via the induction of apoptosis [79].

miRNAs are endogenous ncRNAs that are 19–25 nucleotides long, which bind to the 3′-UTR of target mRNAs to trigger their degradation or inhibit protein translation [63,64,80]. The aberrant expression of miRNAs can play an important role as oncogenes or tumor suppressors in the development and progression of various cancers [81,82,83]. In particular, miRNA expression was significantly altered in various drug-resistant HCC cells compared to the expression in drug-sensitive cells, suggesting that the expression of several miRNAs can contribute to therapeutic effect prediction in HCC [84,85,86,87]. Several miRNAs are associated with resistance to cytotoxic drugs in HCC. In cisplatin-resistant HCC tissues and cells, miR-130a and miR-182 are significantly upregulated, suggesting that these miRNAs are associated with cisplatin resistance [88,89]. MiR-130a targets the tumor suppressor runt-related transcription factor 3 and activates the Wnt/β-catenin pathway. The inhibition of miR-182 can overcome cisplatin resistance in HCC by inhibiting tumor-protein-53-induced nucleoprotein, a tumor suppressor. In contrast, let-7a has been reported to enhance resistance to adriamycin in HCC cell lines [90], as well as miR-519d, which confers adriamycin resistance by targeting tumor suppressor genes such as p21 and PTEN [91]. Conversely, tumor suppressor miRNAs that function as tumor suppressors can overcome adriamycin resistance in HCC, and miR-26a/b has been shown to promote adriamycin sensitivity in HCC cell lines by targeting unc51-like autophagy-activating kinase 1 expression and autophagy [92]. Similarly, miR-520b increases the sensitivity of HCC cells to adriamycin by suppressing the expression of autophagy-related 7, a key autophagy regulator [93]. Although miRNAs also modulate the expression of immune checkpoint molecules in the tumor microenvironment (TME) [94], it is currently unclear how miRNAs are involved in the resistance to immune checkpoint inhibitors (ICIs), an issue that requires further investigation.

## 3. TME and Drug Resistance in HCC

Cancer progression is controlled not only by cancer cells but also by the TME formed by the surrounding non-malignant tumor cells, including lymphocytes, inflammatory cells, endothelial cells, fibroblasts, and mesenchymal stem cells [95]. The TME is involved in tumor formation, survival, metastasis, angiogenesis, fibroblast proliferation, and infiltration of macrophages and other immune cells [96]. Cells in the TME regulate cancer growth through mitogenic and growth-inhibitory signals, and cancer cells produce VEGF, platelet-derived growth factor (PDGF), and colony-stimulating factor 1 to mobilize macrophages, resulting in cell–cell interactions [97]. Fibroblasts are associated with the production of the extracellular matrix, including collagen and fibronectin. The TME is also relevant to the therapeutic targets and the mechanisms of drug resistance.

### 3.1. Vascular System

Several angiogenesis-stimulating factors, such as VEGF, fibroblast growth factor (FGF), PDGF, their corresponding receptors, and endoglin, are associated with HCC growth [98]. Angiogenic molecules are therapeutic targets for HCC and associated with resistance to treatment [99]. Among these molecules, VEGF strongly promotes angiogenesis, and in fact, most of the MTAs approved to date for advanced HCC, such as sorafenib, regorafenib, and lenvatinib, target the VEGF/VEGF receptor (VEGFR) angiogenic pathway. Circulating VEGF levels were shown to be elevated in HCC and correlated with tumor angiogenesis and progression, and an association between high tumor microvessel density and increased local and circulating VEGF with rapid disease progression and poor prognosis [100], supporting the efficacy of targeting the VEGF pathway in HCC therapy. In a report on FGF and VEGF crosstalk, FGF-2 and VEGF-A were associated with increased capillary sinusoids in HCC tumor angiogenesis [101], and FGF stimulation modulated the expression of integrins that regulate endothelial cells in the MTA and alter cell parameters required for angiogenesis. Placental growth factor (PLGF) is a pro-angiogenic factor belonging to the VEGF family, whose overexpression has been observed in several tumor-resistant to anti-angiogenic therapies, making PLGF a potential HCC therapeutic target [102,103].

Angiogenesis inhibitors alone have limited efficacy against HCC due to acquired resistance, which is associated with non-angiogenic mechanisms of tumor vascularization, including vasculogenic mimicry (VM) and vessel co-option (VC). VM is regulated by pluripotent stem-cell-like tumor cells that acquire endothelial-like properties, secrete collagen and proteoglycans to form stable tubular structures without endothelial cells [104], transport nutrients for tumor progression, and are regulated by EMT cancer stem cell properties and the remodeling of the extracellular matrix induced by hypoxia [105]. Blocking VEGF signaling using MTAs inhibits tumor angiogenesis and aids the formation of VM structures by establishing an acidic environment, which is an important contributor to drug resistance. Indeed, a recent study showed that integrin subunit alpha 5 and integrin subunit beta 1 are highly expressed in sorafenib-resistant HCC tissues, promoting hypoxia and VF structure formation [106]. In addition, the molecular mechanisms that promote VM in HCC include the regulation of lysyl oxidase homolog 2 via HIF-1α expression [107], increased translation of yes-associated protein by methyltransferase 3, an m6A methyltransferase [108], activation of the PI3K/Akt/matrix metallopeptidase pathway [109], induction of the EMT pathway by neurogenic locus notch homolog protein 1 [110] and heat shock protein 90 beta (Hsp90β) [111], and induction of VM formation by bone morphogenetic protein 4 [112] and migration-inducing gene 7 [113].

VC is a non-angiogenic mechanism by which cancer cells adhere to normal blood vessels in the original organ and grow invasively along the blood vessels [104]. A study using HCC xenograft models has shown that sorafenib treatment markedly inhibits tumor angiogenesis but preserves VC-associated vasculature, allowing resistant tumors to invade locally, promoting tumor-invasive signals and EMT-like changes that facilitate the transition from angiogenesis to VC [114]. VC is associated with resistance to anti-angiogenic therapy in liver metastases of colorectal cancer, and the expression of molecules related to apoptosis, mobility, and EMT of hepatocytes in the proximity of cancer cells has been found to be upregulated, with cancer cells entering sinusoidal vessels to establish VC [115]. However, whether the same mechanism is relevant to HCC needs further investigation. These mechanisms are also potentially important therapeutic targets for HCC cases that are resistant to anti-angiogenic therapy.

### 3.2. Transport Processes

ATP-binding cassette (ABC) transporters are transmembrane proteins that share a highly conserved ATP-binding domain and use the energy obtained from ATP hydrolysis to actively transport compounds across the cell membrane [116]. ATP-binding cassette subfamily B member 1 (ABCB1) is a multidrug efflux transporter that transports molecules with diverse chemical structures out of the cells and is highly expressed in the apical membrane of many tissues involved with pharmacokinetics, including the small intestine, blood–brain barrier, liver, kidney, and genitalia. Currently used anticancer drugs and tyrosine kinase inhibitors (TKIs) include substrates of ABC transporters and are important contributors to drug resistance [117]. In particular, sorafenib, which is frequently used in the treatment of advanced HCC, has been shown to cause drug resistance by interacting with ABC transporters. ABC transporters have been implicated in sorafenib resistance by reducing the accumulation of the drug in HCC cells via active efflux [117]. Based on these findings, it appears that TKIs act as either organopathic or inhibitory agents depending on the expression of specific pumps, the type of drugs used in combination, their dosage, and their affinity for ABC transport.

Exosomes, which are involved in intracellular communication, also act as transporters in vivo and are involved in drug delivery [118]. In normal cells, exosomes transport harmful biological substances, but in cancer cells, this transportation may be inhibited. Several studies have shown that in drug-resistant cancer cells, therapeutic drugs are encapsulated in exosomes and transported out of the cancer cells [118,119]. Exosomes and their contents have been shown to be potential therapeutic targets for a variety of cancer types, and the involvement and functional roles of ncRNAs in exosomes have been elucidated [120]. LincRNA VLDLR (linc-VLDLR) is significantly upregulated in HCC. Exposure to various therapeutic agents such as camptothecin, doxorubicin, and sorafenib increases linc-VLDLR expression in exosomes released from HCC and is associated with drug resistance. The knockdown of linc-VLDLR inhibits HCC growth, suggesting that linc-VLDLR might be a potential therapeutic target. Therapeutic targeting of miRNAs in exosomes has been shown to promote sensitivity to chemotherapy in HCC [121]. Exosomes extracted after the transfection of adipose-derived mesenchymal cells with miR-122 and added to HHC cells enhance the sensitivity of these exosome-treated HHC cells to chemotherapeutic agents such as sorafenib.

### 3.3. Immune System

The liver is exposed to a variety of biological substances and drugs in the body through the intestinal circulation and is equipped with an anti-inflammatory immune environment by Kupffer cells (KCs), hepatic stellate cells (HSCs), and liver sinusoidal endothelial cells (LSECs). KCs are resident liver macrophages that act as antigen-presenting cells along with LSECs and HSCs. Further, immune-associated cells in the TME play a crucial role in the progression, metastasis, and drug resistance of HCC cells. An imbalance in the tumor-immune microenvironment, consisting of immune suppressor cells, including myeloid-derived suppressor cells (MDSCs), tumor-associated macrophages (TAMs), regulatory T cells (Tregs), and anti-tumor effector cells such as cytotoxic T lymphocytes (CTLs), natural killer (NK) cells, and dendritic cells, leads to resistance to immunotherapy [122,123]. Recently, the combination of tremelimumab, anti-cytotoxic T-lymphocyte-associated antigen-4 antibody (CTLA-4) and durvalumab, and anti-programmed cell death-ligand 1 (PD-L1) antibody has been shown to have clinical activity and safety in a Phase II trial of uHCC [124]. Furthermore, in a Phase III trial, single tremelimumab regular interval durvalumab regimen demonstrated superior overall survival to sorafenib in patients with uHCC and no previous systemic treatment [125]. Combination immunotherapy with anti-CTLA-4 and anti-PD-L1 antibody appears to be a promising therapeutic strategy for uHCC in the future.

MDSCs are heterogeneous immature cells originating from bone marrows that are mobilized in local tissues and blood in various situations, including tumors, infections, autoimmune diseases, and trauma, and play an important role in suppressing anti-tumor immunity in cancer-bearing hosts [126,127]. MDSCs may consume excessive amounts of essential amino acids, such as arginine 1 and cysteine, as well as tryptophan, due to the overexpression of indoleamine-pyrrole 2,3-dioxygenase in the body, leading to T cell dysfunction [128,129]. Deficiencies in these amino acids downregulate T receptor activity on CD8-positive T cells in the liver, allowing tumor cells to evade immune response [130]. In another report, MDSCs have been shown to suppress cancer immune function by producing reactive oxygen and nitrogen species, promoting antigen-specific T-cell tolerance, and inhibiting T-cell migration to the tumor [131]. More importantly, cytokines produced by tumor cells, such as VEGF, FGF, HIF-1α, and interleukin 6 (IL-6), have been shown to promote MDSC accumulation and are associated with therapeutic resistance [132,133,134,135]. A previous study has revealed that chemotherapy-resistant HCC, such as 5-FU, increases MDSC activity, and that the use of the anti-IL-6 neutralizing antibody in combination with 5-FU significantly reduces tumor growth and can help overcome drug resistance [135]. The inhibition of cell-cycle-related kinases, which are intrinsic to HCC, has also been shown to weaken cancer immunosuppression by MDSCs, thus further enhancing the blocking effect of PD-L1, making it a potential therapeutic target [136].

TAMs that accumulate in tumors include tissue-resident macrophages, monocyte-derived exudate macrophages, and monocyte-derived exudate macrophages mobilized by inflammatory stimuli. As tumor size increases, monocyte migration to the tumor stroma and differentiation into TAMs are induced by TAMs, and exudate TAMs become the main components [137,138]. The classification of macrophages by activation state divides them into two subtypes: M1, tumor suppressor macrophages that undergo classical activation by interferon alpha (IFN-α), IFN-β, and IFN-γ, an indicator of the Th1 response, and M2, tumor-promoting macrophages that are activated by anti-inflammatory factors such as Th2, IL-4, and IL-10 [96]. Although the M1/M2 balance varies by cancer type, M2 is predominant in HCC [139], indicating that M2 TAM infiltration is associated with poor prognosis and therapeutic resistance in various cancers [140,141]. In HCC, HGF derived from M2 TAMs accumulates in HCC cells and activates HGF/c-Met, extracellular signal-regulated kinase 1 (ERK1)/ERK2/mitogen-activated protein kinase (MAPK), and PI3K/Akt signals through a feed-forward control, which in turn attracts M2 TAMs. Ultimately, this process is associated with tumor growth and drug resistance [142]. In another study, TAMs have been shown to induce oxaliplatin resistance via autophagy in HCC cells; hence, inhibiting TAM recall and altering TAM polarity may be a novel therapeutic strategy to increase treatment sensitivity against HCC [143].

Tregs are a subset of CD4+ T cells that are induced by transcription factors such as FOXP3 and are responsible for autoimmune tolerance and homeostasis, suppressing the excessive immune response that causes allergic and autoimmune diseases, and promoting tumor progression [96]. The number of Tregs increases in the tumors and blood of HCC patients compared to that in healthy controls, and the percentage and absolute number of CD4+ CD25+ T cells are significantly increased in the tumor periphery [144]. As immune checkpoint molecules such as T-cell immunoglobulin mucin 3 emerge from Tregs and block the activation of effector T cells, high Treg expression has been reported to be associated with poor prognosis in HCC patients following programmed cell death 1 (PD-1)/PD-L1 therapy [145]. Furthermore, transforming growth factor beta (TGF-β) secreted by Tregs induces EMT, indicating that the use of a TGF-β inhibitor contributes to the sensitivity of HCC cells to sorafenib and regorafenib [146]. C-C chemokine receptor type 4 (CCR4)-positive Tregs, which are recruited following HBV and HCV infection and are also associated with resistance to sorafenib, display a higher expression of IL-10 and IL-35 and are more suppressive of CD8+ T cells [147,148]. Treatment with a CCR4 antagonist blocks Treg accumulation in HCC tumors, overcoming sorafenib resistance, and sensitizing tumors to PD-1 inhibition.

NK cells are a component of the innate immune system and act as the most important defense against cancer cell invasion by regulating cytotoxicity and cytokine production. Liver-specific NK cells notably have the strongest NK activity in any organ [149,150]. TME-induced NK-cell abnormalities are the main mechanism by which cancer cells evade tumor-immune responses. Immunosuppressive factors, such as TGF-β, IL-6, IL-10, and IL-23 secreted by MDSCs, TAMs, and Tregs, suppress NK cell function, causing cancer-immune evasion and promoting tumor progression [151]. In addition, several recent studies have reported that the susceptibility of NK cells is regulated by various factors influencing HCC and TME, such as the inhibition of the CCR4-Not transcription complex subunit 7, and the enhancement of EZH2. Granulin-epithelin precursor and miR-889 can cancel NK cell resistance, indicating that these molecules are promising targets in cancer immunotherapy for HCC patients [152,153,154,155].

Thus, the immune microenvironment of HCC is rich in inflammatory chemokines, cytokines, and immunosuppressive molecules that create a strongly immunosuppressive tumor environment and play an important role in reorganizing TME, mediating intercellular cross-talk, and promoting immune evasion in HCC. The most studied immune checkpoints in HCC are CTLA-4, PD-1, PD-L1, and mucin-domain-containing molecule 3 (Tim-3). CTLA-4 is an inhibitory co-receptor constitutively present on Tregs, which plays an important role in regulating CD4+ T cell function, and in HCC, as in other cancer types, it inhibits T cell proliferation through the recognition and differentiation of tumor-associated antigens [156]. Furthermore, in HCC tissues, CTLA-4 contributes to tumor growth by promoting immunosuppression through the induction of Treg activity and the production of indoleamine-2,3-dioxygenase and IL-10 in DCs [157]. PD-1 is a regulatory immunoglobulin expressed on activated CD4+ and CD8+ T cells, B cells, and NK cells and plays an important role in maintaining immune tolerance and suppressing T lymphocyte cytotoxicity [158]. It is also known that the upregulation of PD-L1 on HCC cells induced by various cytokines, especially IFN-γ, contributes to the impairment of anti-tumor immunity and promotes the apoptosis of CD8+ T cells [159]. Tim-3 is a transmembrane immunoglobulin expressed on IFN-γ-secreting Th1 cells, NK cells. Tim-3 expression is increased in T cells infiltrating chronic HBV infection, and the Tim-3/galectin-9 pathway is associated with poor prognosis in patients with HBV-associated HCC [160]. The clinical value of these immune checkpoint molecules in HCC needs to be further elucidated.

## 4. Sorafenib Drug Resistance in HCC

Sorafenib was the first MTA to show a survival benefit in patients with advanced HCC. The Asia-Pacific trial and SHARPP trial in patients with Child–Pugh Class A and advanced HCC have previously shown longer median OS and time to progression with sorafenib treatment than with placebo [9,161]. Robust clinical studies on sorafenib have continued ever since, and despite an increase in the number of available MTAs for treatment use, sorafenib remains the most empirically supported treatment drug, with the most evidence available regarding the mechanisms of drug resistance. Sorafenib inhibits tumor growth in HCC by blocking Raf-1, B-Raf, and Ras/Raf/mitogen-activated protein kinase(MEK)/ERK signaling kinase activity, as well as by inhibiting the angiogenesis-targeting PDGFR-β, VEGFR2, and cKIT [162].

Epigenetic modifications by DNA methylation and ncRNA in HCC, in addition to cell proliferation and differentiation, are closely related to the mechanisms of sorafenib resistance (Table 1). A study analyzing global methylation in HCC cells treated with sorafenib has revealed the tendency of oncogenes and tumor suppressor genes to be hypermethylated and hypomethylated, respectively, following sorafenib treatment [163]. In addition, genes with varying degrees of methylation include those associated with apoptosis, invasion, and angiogenesis, as well as genes related to pathways known to be downregulated in HCC, including RAF/MEK/ERK, Janus kinase-STAT, PI3K/Akt/mammalian target of rapamycin (mTOR), and nuclear factor-kappa B (NF-κB). A study using HCC cell lines and xenograft models has reported that microrchidia 2 (MORC), which forms a complex with DNMT3A on the promoters of neurofibromatosis type 2 (NF2) and kidney and brain protein (KIBRA) that cause DNA hypermethylation and transcriptional repression, is associated with HCC resistance to sorafenib and maintenance of oncogenic potential [164]. Another study showed that the methylation of the promoter region of the H19 gene is associated with sorafenib resistance in HCC cell lines and that the overexpression of H19 sensitizes sorafenib resistance by suppressing cell growth after sorafenib treatment [165]. Furthermore, protein arginine N-methyltransferase 6 has been shown to methylate arginine 100 at CRAF and inhibit the former extracellular matrix complex subunit/RAF binding ability, thereby altering the ERK-mediated nuclear transport of pyruvate kinase M2 isoform (PKM2) and reducing sorafenib resistance in HCC cells [166]. The overexpression of HIF-1α and HIF-2α as well as the methylation-dependent knockdown of Bcl-2 interacting protein 3 (BNIP3) are associated with the development of sorafenib resistance in HCC, and the demethylation of the BNIP3 promoter can restore BNIP3 expression, suggesting its potential as a molecular target to overcome sorafenib resistance [167].

In recent years, there has been mounting evidence of ncRNA-based mechanisms, including those involving lncRNAs and miRNAs, being associated with sorafenib resistance in HCC. A study using sorafenib-resistant HCC cells has shown that sorafenib reduces miR-21 expression in the nucleus and promotes lncRNA small nucleolar RNA host gene 1 (SNHG1) expression, leading to the activation of the AKT pathway, contributing to drug resistance [168]. The lncRNA SNHG3 also causes EMT in HCC cells through miR-128/CD151 cascade activation and is involved in sorafenib resistance [169]. Furthermore, SNHG16 expression is upregulated in HCC cell lines and tissues, and even more so in sorafenib-resistant HCC cells, suggesting that SNHG16 knockdown can improve sensitivity to sorafenib-resistant HCC cells in vitro and in vivo [170]. SNHG16 functions as an endogenous sponge for miR-140-5p in HCC cells, and the overexpression of miR-140 increases the sensitivity of sorafenib-resistant HCC cells to sorafenib. Additionally, the effect of the SNHG16 knockdown on sorafenib resistance can be blocked by an miR-140-5p inhibitor.

In a study investigating the role of the lncRNA nuclear-enriched abundant transcript 1 (NEAT1) in regulating HCC sensitivity to sorafenib, miR-335 has been found to increase sorafenib sensitivity via apoptosis induction and reduce tumor size in xenograft mouse models implanted with HCC cells following the knockdown of the NEAT1 gene [171]. miR-335 is negatively regulated by NEAT1 and is associated with sorafenib resistance in HCC by inhibiting the cMet-AKT pathway. Another study on the role of forkhead box protein D2-antisense RNA 1 (FOXD2-AS1) showed that in sorafenib-resistant HCC cells, FOXD-AS1 can function as a sponge for miR-150-5p, significantly decreasing transmembrane protein 9 (TMEM9) and increasing miR-150-5p expression [172]. The overexpression of FOXD2-AS1 can overcome drug resistance in sorafenib-resistant HCC cells by increasing TMEM9 expression, whereas the knockdown of FOXD2-AS1 decreases TMEM9 expression and increases sensitivity to sorafenib in HCC cells, suggesting that FOXD2 could serve as a therapeutic target for HCC.

The association between miRNAs and sorafenib resistance has also been studied extensively. miR-622 contributes to the suppression of drug resistance in sorafenib-resistant HCC cells by targeting KRAS and inhibiting RAF/ERK and PI3K/Akt signaling [177]. Clinically, the expression of heterogeneous nuclear ribonucleoprotein A1 (hnRNPA1) and PKM2 has been reported to be upregulated in patients with sorafenib-resistant HCC and is inversely correlated with the expression level of miR-374b [175]. An experiment using sorafenib-resistant HCC cells has revealed that miR-374b binds to the 3′-UTR of hnRNPA1 and downregulates its expression, resulting in reduced PKM2 levels, suggesting that the miR-374b/hnRNPA1/PKM2 axis is an important mechanism in acquiring sorafenib resistance in HCC cells. Several studies on other miRNAs have shown that the aberrant expression of miR-19a-3p induces sorafenib resistance in HCC cells by regulating the PTEN/Akt pathway [86], and that the overexpression of miR-494 enhances sorafenib resistance against HCC cells via the activation of the mTOR pathway [176]. mir-221, which is aberrantly expressed in HCC, exerts its anti-apoptotic activity by targeting caspase-3, and is involved in sorafenib resistance in HCC [174]. Another study suggested that the sorafenib treatment of HCC cells increases the expression of the pro-apoptotic factor p53-upregulated modulator of apoptosis and activates poly-ADP-ribose polymerase and caspase-3, and that miR-181a induces sorafenib resistance via the suppression of Ras association domain-containing protein 1 [173]. These data provide important insights into promising therapeutic strategies to overcome sorafenib resistance by targeting ncRNAs.

In the sorafenib treatment of HCC, the anti-angiogenic effect arises via the inhibition of HIF-1α synthesis and the attenuation of VEGF expression [178]. Continued sorafenib treatment inhibits tumor angiogenesis, resulting in a hypoxic environment within the tumor and facilitating the selection of resistant cell clones that attempt to adapt to oxygen and nutrient deprivation, leading to sorafenib resistance via the activation of HIF-1α and NF-κB in HCC [179]. EF24, a structurally similar substance to curcumin, inhibits HIF-1α and promotes its degradation by upregulating the von Hippel–Lindau tumor suppressor, synergistically enhancing the anti-tumor effects of sorafenib and abrogating drug resistance. The overexpression of HIF-2α, similar to that of HIF-1α, has also been shown to be associated with poor prognosis in patients with HCC [180], and the sorafenib-induced upregulation of HIF-2α is associated with drug resistance through the activation of the TGFα/epidermal growth factor receptor (EGFR) pathway [181]. Furthermore, the HIF-2α inhibitor increases the androgen receptor and suppresses the STAT3/pAkt/pERK pathway, thereby enhancing sorafenib sensitivity [182]. These studies suggest that hypoxia affects sorafenib treatment and that high HIF expression leads to sorafenib resistance, further implying that the inhibition of HIF could be an effective approach to suppressing drug resistance.

Similar to other drugs, sorafenib resistance in HCC involves ABC transporters, which withdraw the drug from HCC cells, thereby reducing its anti-tumor effect [117]. Sorafenib is also encapsulated in exosomes and transported out of HCC cells, enhancing drug resistance [118]. Recently, several TKIs, including sorafenib, have been reported to interact with the ABC transporters ABB1, ABCC1, ABG2, and ABCC10, and function in a complex manner as either organotypic or inhibitory agents, depending on specific pump expression, drug concentration, affinity for the transporters, and the type of drug in combination [117]. The natural sesquiterpene components of many essential oils inhibit the ABC pump and can increase the sensitivity of HCC cells to dosages of sorafenib that do not exhibit anti-tumor effects. They also enhance the cytotoxic response by inhibiting sorafenib degradation and promoting its intracellular accumulation [183]. Interestingly, another study has shown that COP9-signaling corset 5 (CSN5) is associated with sorafenib resistance in HCC cells, and that silencing CSN5 expression abrogates resistance to sorafenib and downregulates ABCB1, ABCC2, and ABCG2 [184]. Studies using resistant HCC cells established via continuous culture at gradually increasing sorafenib concentrations showed that multidrug-resistance-associated protein 3 (MRP3), an efflux transporter involved in multidrug resistance, is expressed at higher levels in resistant HCC cells than in parent cells [185]. MRP3 knockdown in sorafenib-resistant HCC cells restores sensitivity to sorafenib. In contrast, studies on exosomes showed that exposure of HCC cells to sorafenib increases the expression of linc VLDLRs in exosomes related to the cells, and incubation with these exosomes inhibits cell death in response to the drug, and increases linc VLDLR expression, leading to drug resistance [120]. miRNAs are also transported in exosomes and are associated with HCC-related sorafenib sensitivity mediated by exosomal miR-122 secreted from adipose tissue-derived mesenchymal stem cells transfected with miR-122 [121]. In vivo experiments also showed that the intratumoral injection of exosomal miR-122 can markedly attenuate sorafenib resistance. In another study, a combination of sorafenib and exosomes modified with glucose-regulated protein 78 (GRP78) sensitizes sorafenib-resistant HCC cells and overcomes drug resistance by targeting GRP18 in HCC cells [186]. Loading exosomes with functional proteins, ncRNAs, or therapeutic drugs could be an effective therapeutic strategy.

Immunocompetent cells in the TME have been shown to play an important role in the acquisition of sorafenib resistance in HCC. TAMs suppress anti-tumor immunity by expressing cytokines and chemokines; promoting phagocytosis or inhibiting the proliferation of TAMs are potential strategies to overcome resistance. A natural product from *Abies sachalinensis* has been shown to be potent against HCC cells and can improve the therapeutic effect of low-dose sorafenib by increasing the number of intratumoral CD8+ T cells and enhancing tumor cell death [187]. In orthotopic mouse models of HCC, sorafenib increases the number of F4/80+ TAMs, as well as CD11b + Gr-1+ and CD45+ CXC motif chemokine receptor 4 (CXCR4)+ myeloid cells [188]. Furthermore, sorafenib treatment increases the number of CD4 + CD25 + FOXP3+ Treg infiltrating HCC and inhibits CXCR4, preventing drug resistance due to the immunosuppressive microenvironment established following sorafenib treatment, suppressing tumor growth. Macrophages secrete HGF accompanied by a significant increase in M2 over M1 type macrophages, which has been shown in vitro and in vivo to significantly increase resistance to sorafenib by maintaining tumor growth [142]. As for tumor-associated neutrophils (TANs), which regulate cancer progression through the release of cytokines, the regulatory mechanisms in HCC are less clear, but surgically resected HCC tissues with preoperative sorafenib treatment contain more TANs than those without prior treatment [189]. In addition, sorafenib has been shown to increase the number of TANs in tumors and the expression of the C-C motif ligand 2 (CCL2) and CCL17 in mouse models of HCC-bearing carcinomas. Another study provides important evidence regarding the expression level of pERK, a candidate marker for predicting response to sorafenib treatment in HCC. pERK-expressing HCC tissues show a marked increase in CD8 + CTLs in the tumor and a high PD-1 expression, suggesting that anti-PD-1 therapy could overcome sorafenib resistance in HCC [190]. Immune combination therapies, such as the atezo + bev combination for advanced HCC, are currently in clinical use, and the modification of the TME with ICIs could be an important approach for sorafenib-resistant HCC.

## 5. Regorafenib Drug Resistance in HCC

Regorafenib is an effective second-line treatment for HCC progression. In the RESORCE trial, the median OS of patients treated with regorafenib after sorafenib treatment was 10.6 months, compared with 7.8 months for the placebo group, and the median progression-free survival (PFS) was 3.1 months for the regorafenib-treated group and 1.5 months for the placebo group, both showing a significant increase with regorafenib treatment [10]. Regorafenib is an MTA that targets VEGFR1-3 and PDGFRA, inhibits receptor tyrosine kinases, such as KIT and RET, and exhibits higher efficacy in STAT3 inhibition [191]. Furthermore, regorafenib restores sorafenib sensitivity by inhibiting ERK and STAT3 in HCC cells that acquire sorafenib resistance through HGF stimulation [192]. FOXO3 is also associated with sorafenib resistance in HCC through the overactivation of autophagy, but regorafenib is able to inhibit this regulatory mechanism and improve therapeutic efficacy [193].

A recent study on the cancer immune environment showed that regorafenib inhibits STAT3 and increases the expression of CXCL10, a ligand for CXCR3 expressed on tumor-infiltrating T lymphocytes, thereby promoting the infiltration of CTLs into tumors [194]. Regorafenib also interferes with EMT progression by inhibiting ERK/STAT3 signaling [192]. Several studies on the predictors of therapeutic response to regorafenib in patients with HCC have reported that protein levels in the patients’ serum, including lectin-like oxidized low-density lipoprotein receptor 1 [195] associated with hypoxia-induced TAMs, angiopoietin 1, which promotes angiogenesis, and annexin A3 [196], which is associated with apoptosis, were correlated with OS. miRNAs have also been shown to be prognostically useful, with the levels of 9 miRNAs, namely miR-15b, miR-30a, miR-107, miR-122, miR-125b, miR-200a, miR-320, miR-374b, and miR-645, in the plasma of regorafenib-treated HCC patients correlating with OS [195].

Various TMEs related to hypoxia, EMT, cell cycle, and apoptosis are also intricately involved in HCC resistance to regorafenib, as shown in Table 2. EMT appears to be a central mechanism in the emergence of regorafenib resistance to HCC, and regorafenib-resistant HCC cells overexpress peptidyl-prolyl cis-trans isomerase 1 (Pin1), which regulates the expression of EMT-related molecules such as E-cadherin and promotes HCC progression, invasion, and metastasis [197]. Elevated TNFα expression promotes HCC resistance to sorafenib in vitro by inducing EMT [198], and it is also associated with EMT in regorafenib resistance. FOXM1, a transcription factor for cell-cycle-associated molecules, regulates cancer stem cells and is associated with resistance to chemotherapy. Elevated FOXM1 expression is also associated with regorafenib resistance and decreases survival in patients with HCC [199]. As a mechanism associated with apoptosis, the high expression of topoisomerase 2 alpha (TOP2A), which has been clinically correlated with poor prognosis, is also involved in the resistance to regorafenib [200]. Sustained exposure to regorafenib elevates TOP2A, and conversely, the suppression of TOP2A improves regorafenib sensitivity. A different resistance mechanism involves the activation of Wnt/β-catenin signaling, which can protect HCC cells from regorafenib-induced apoptosis. TGF-β signaling activity is markedly elevated in resistant cells established for long-term regorafenib treatment, and they can be rendered sensitive to regorafenib again by inhibiting the TGF-β pathway [148]. A previous study using clustered regularly interspaced short palindromic repeats/CRISPR-associated protein 9 (CRISPR/Cas9) activation has demonstrated that hexokinase 1 (HK1), which catalyzes glucose metabolism, confers regorafenib resistance, and could serve as a biomarker and therapeutic target [201]. Activating transcription factor 3 (ATF3)-mediated upregulation of the interleukin-6 receptor alpha (IL-6Rα) can induce multifunctional cytokines and promote resistance to regorafenib and sorafenib [202]. Regorafenib-resistant HCC cells show a high expression of sphingosine kinase 2 (SphK2), indicating that SphK2/sphingosine-1-phosphate (S1P) mediates regorafenib resistance via the activation of NF-κB and STAT3. Importantly, the knockdown of SphK2 restores HCC cell susceptibility to regorafenib [203]. Hypoxia associated with angiogenesis, led by a high dose of multi-kinase inhibitors, including regorafenib, has been suggested to induce the immunosuppression of the TME [204]. Therefore, combining regorafenib with ICIs may be a novel therapeutic strategy to overcome regorafenib resistance, as the combination of ICI and VEGF inhibitors, such as atezo + bev combination therapy, has shown anti-tumor synergistic effects against HCC [205].

## 6. Lenvatinib Drug Resistance in HCC

Lenvatinib is a multi-kinase inhibitor targeting VEGFR1-3, FGFR1-4, PDGFRA, and tyrosine kinase receptors, and its inhibitory effect on VEGFR and FGFR is stronger than that of sorafenib [206]. Lenvatinib, like other MTAs, inhibits angiogenesis and alters TME. In the REFLECT trial, patients with advanced HCC treated with lenvatinib exhibited an OS of 13.6 months, compared with 12.3 months with sorafenib treatment, indicating a similar efficacy [11]. In addition, the anti-tumor effect indicates that lenvatinib is significantly superior to sorafenib in terms of response rate, disease control rate, and PFS.

Several studies have investigated factors that predict the efficacy and outcome of the lenvatinib treatment. Clinically, knockdown of muskelin 1 antisense RNA (MKLN1-AS), a lncRNA that is associated with vascular invasion and poor prognosis and is upregulated in HCC tissues, enhances the pro-apoptotic effects of lenvatinib, and thus, could be used as a therapeutic target [207]. miR-3154 directly targets HNF4α, which is associated with tumorigenesis, growth, and metastasis in HCC and has been shown to be a potential predictor of clinical response to lenvatinib [208]. In another report on lenvatinib sensitivity, HCC cells became sensitive to lenvatinib when the expression of nuclear factor erythroid-derived 2-like 2 (Nrf2) was silenced. Nrf2 protects against ferroptosis, a type of cell death caused by the iron-dependent accumulation of lipid-reactive oxygen species, and HCC cells expressing Nrf2 have been found to be resistant to lenvatinib. Lenvatinib is also less effective against HCC lesions with low FGFR4 expression [209]. Another study found that nucleotide-binding oligomerization domain 2 (NOD2), an innate immune sensor that elicits a strong immune response against pathogens, significantly enhances the sensitivity of HCC cells to various therapeutic agents, including lenvatinib, through the AMPK signaling pathway [210].

Various TMEs, such as epigenetic regulation, transport processes, hypoxia, and autophagy are also intricately involved in lenvatinib resistance in HCC, as shown in Table 3. Histone modifications and specific ncRNAs are associated with lenvatinib resistance; an RNA sequencing study discovered that the lncRNA X-inactive specific transcript (XIST) is highly expressed in lenvatinib-resistant HCC cells and that lncXIST promotes lenvatinib resistance via the activation of the EZH2/NOD2/ERK-signaling axis [211]. The recently reported lncRNA AC026401.3 is upregulated in HCC tissues and correlates with advanced stages and poor prognosis in HCC patients, and the knockout of AC026401.3 enhances the sensitivity of HCC cells to lenvatinib. AC026401.3 interacts with the organic cation uptake transporter 1 (OCT1) to activate the E2F2 promoter region and induce the transcription of E2F2, consequently enhancing lenvatinib resistance [212]. Several miRNAs have been shown to be altered in lenvatinib-resistant HCC, and the downregulation of miR-128-3p displays the strongest activity in negatively regulating c-Met, as it is involved in the resistance mechanism via AKT, which regulates the apoptotic pathway, and ERK, which regulates the cell cycle [213]. Among the circulating ncRNAs, circulating p2 of 1-activated kinase 1 (circPAK1) is overexpressed in HCC cell lines and tissues and correlates with poor prognosis in HCC patients. circPAK1 can be transported via exosomes from lenvatinib-resistant cells to lenvatinib-sensitive cells and can promote lenvatinib resistance in the receiving cells [214]. CircRNA mediator complex subunit 27 (circMED27) has also been shown to be significantly upregulated in the serum of patients with HCC, correlating with poor clinical features and prognosis. As an endogenous RNA competing with miR-655-3p, it upregulates the expression of ubiquitin-specific peptidase 28 (USP28) to promote lenvatinib resistance [215].

Based on a previous report on transport processes, the expression of MRD1 and breast cancer resistance protein (BCRP) transporter is known to be markedly elevated in lenvatinib-resistant HCC cells, and the activation of EGFR, MEK/ERK, and PI3K/AKT signaling is associated with the acquisition of drug resistance [216]. Combining lenvatinib with elacridar, a dual inhibitor of MDR1 and BCRP, or gefitinib, an EGFR inhibitor, may improve drug sensitivity in lenvatinib-resistant HCC and represent a viable therapeutic strategy. Regarding exocytosis, a form of extracellular secretion, lenvatinib exocytosis can markedly enhance lenvatinib exocytosis by activating EGFR and STAT3/ABCB1 signaling, leading to lenvatinib resistance [217]. Neuropilin-1 (NRP1), a co-receptor associated with chemotherapy resistance, is also associated with lenvatinib resistance in HCC in terms of hypoxia and autophagy interactions [218]. Lenvatinib suppresses autophagy by inhibiting NRP1 expression in HCC cells; however, although co-treatment with bafilomycin A1 reduces the anti-tumor effect of lenvatinib, silencing the NRP1 gene can also reduce the efficacy of lenvatinib even in the presence of bafilomycin A1. Furthermore, HIF1α directly regulates NRP1 expression in cells exposed to a hypoxic microenvironment, suggesting a role for HIF1α-induced hypoxic responses in promoting lenvatinib resistance, particularly when the silencing of HIF1α enhances the anti-tumor effects of lenvatinib. Lenvatinib indirectly suppresses fibronectin in HCC cells under normoxic condition in vitro, but hypoxia induces transcription factors including HIF-1α, which increase fibronectin expression leading to the lenvatinib resistance in HCC cells under hypoxic conditions [219].

In our previous study, in which HCC cells were exposed to lenvatinib and cultured long-term to establish resistant HCC cells, the activation of ERK1 signaling was observed in all three lenvatinib-resistant HCC cell lines, including Huh-7, Hep3B, and Li-7, and was associated with resistance mechanisms [220]. Furthermore, cisplatin shows an effective anti-tumor effect on lenvatinib-resistant HCC cells via the ATM/ATR–Chk1/Chk2-signaling pathways. In another report using lenvatinib-resistant HCC cells, the activation of MAPK/MEK/ERK signaling and the increase in the expression of EMT markers were observed, indicating a higher proliferative and invasive potential [221]. Cytokines involved in angiogenesis, including VEGF, PDGF-AA, and angiogenin, were also evaluated in lenvatinib-resistant HCC cells. In contrast, another study has shown that VEGFR2 expression and its downstream Ras/MEK/ERK signaling are elevated in lenvatinib-resistant HCC cells, with no changes in VEGFR1, VEGFR3, FGFR1-4, and PDGFRα/β expression [222]. Furthermore, ETS proto-oncogene 1 is involved in lenvatinib resistance mediated by VEGFR2, and the alkaloid extract sophoridine can overcome lenvatinib resistance by inhibiting proliferation, colony formation, and cell migration. A study using the genome-wide CRISPR/Cas9 library screening system has also identified six genes, including dual specificity phosphatase 4 (DUSP4), as the genes associated with lenvatinib resistance, and the knockout of DUSP4 promotes cell proliferation and migration of HCC during lenvatinib treatment and activates ERK/MEK signaling at the phosphorylation level, inducing lenvatinib resistance [223]. Recent reports support the hypothesis that the activation of ERK signaling is an important mechanism of lenvatinib resistance in HCC.

## 7. Resistance to Other Drugs in HCC

While TKIs, including sorafenib and lenvatinib, have been the first-line treatment for advanced HCC, regimens that include ICIs have recently entered clinical practice. In 2020, the IMbrave150 study in patients with unresectable HCC and no prior systemic drug therapy showed a statistically significant difference in OS between sorafenib treatment and atezo + bev treatment, a combination therapy consisting of an anti-PD1 inhibitor and an anti-VEGF inhibitor [14]. The 6-month and 1-year OS rates were 84.8% and 67.2% in the atezo + bev group and 72.2% and 54.6% in the sorafenib group, respectively, indicating the superior outcome of the atezo + bev combination therapy. PD-L1 expression has been shown to correlate with ICI benefits in several cancer types [224,225], and intertumoral PD-L1 expression is associated with response in patients with HCC who were treated with nivolumab [226]. However, an integrated molecular analysis of 358 baseline tumors from HCC patients enrolled in the GO30140 Phase 1b and IMbrave 150 Phase III trials and treated with atezo + bev, with atezolizumab alone, or with sorafenib has shown that PD-L1 expression did not significantly correlate with improved clinical outcomes following the atezo + bev combination therapy, yet a high PD-L1 mRNA expression and Teff gene signature, as well as a high density of CD8+ T cells and enriched inflammatory response pathways remain associated with the benefit of the atezo + bev combination therapy [227]. Conversely, a high Treg/Teff ratio and a high expression of AFP and GPC3 are associated with a lower efficacy of the atezo + bev combination therapy. In addition, patients with telomerase reverse transcriptase (TERT) promoter mutations exhibited better clinical outcomes with atezo + bev than patients with a non-mutated copy did, and with sorafenib treatment, the presence of TERT promoter mutations is not associated with treatment response or prognosis. The presence or absence of the catenin beta 1 (CTNNB1) mutation does not affect the benefit of the atezo + bev combination therapy, but with sorafenib treatment, patients with CTNNB1 mutation displayed a better prognosis than patients with the wild-type form of the gene did. Although about 25% of HCC tumors have actionable mutations, the prevalence of most mutations is less than 10% [228,229]. The dominant mutation drivers in HCC, such as TERT and CTNNB1, appear to be feasible therapeutic targets. The research on the prediction of response and resistance to the atezo + bev combination therapy in HCC remains limited, and future exploration of therapeutic strategies to overcome resistance to cancer immunotherapy is needed.

Cabozantinib is a multi-kinase inhibitor that targets Met, RET, VEGFR1-3, and AXL receptor tyrosine kinases. In the Phase III CELESTAL trial, enrolled patients with advanced HCC, previously treated with other MTAs, had a median OS and PFS of 10.2 and 5.2 months for the cabozantinib treatment and 8.0 and 1.9 months for the placebo treatment, respectively, showing a significantly improved prognosis [12]. Cabozantinib is also effective in advanced renal cell carcinoma and medullary thyroid carcinoma, and the promotion of angiogenesis involving the Met pathway is resistant to VEGFR [230]. The dual inhibition of VEGFR and c-Met together with the cabozantinib treatment, could be important in overcoming resistance in HCC. In renal cell carcinoma, tumors overexpressing Met show a better response to cabozantinib than Met-negative tumors do [231], but it is unclear whether the degree of Met expression in HCC is associated with the therapeutic effect of cabozantinib, therefore, requiring further investigation.

Ramucirumab is an IgG1 monoclonal antibody that inhibits VEGFR2. Although the REACH trial in patients with sorafenib-refractory advanced HCC, ramucirumab treatment as second-line therapy failed to demonstrate superiority over best supportive care [232]. Yet, the Phase III REACH-2 trial, which focused on patients with AFP ≥ 400 ng/mL, has shown that the median OS and PFS in the ramucirumab-treated group were 8.5 and 2.8 months, respectively, whereas the median OS and PFS in the placebo group were 7.3 and 1.6 months, respectively. This indicates that ramucirumab elicits superior clinical outcomes [13]. As with bevacizumab and sorafenib, resistance to anti-VEGF therapy with ramucirumab is often thought to result from an escape mechanism of the angiogenic process through the activation of pathways other than the VEGF pathway [233], although the evidence according to the mechanisms of resistance acquisition for ramucirumab in HCC and other cancer types is limited. As more systemic therapeutic options become available, the elucidation of cancer resistance mechanisms to cabozantinib and ramucirumab, which may be administered in end-stage treatment lines, will become more vital to improve treatment strategies for advanced HCC.

In addition, in the first regimen that does not include MTAs, tremelimumab plus durvalumab combination immunotherapy, has shown safety and efficacy in uHCC in Phase II and III trials [124,125]. Furthermore, several clinical trials have been conducted on ICI-based therapies in later-line treatment, such as nivolumab [234], nivolumab plus ipilimumab [235], and pembrolizumab [236], and the results are promising for overcoming the TME that leads to drug resistance in previously used MTAs.

## 8. Conclusions

Although several MTAs and ICIs are in clinical use for systemic therapy of advanced HCC, the therapeutic outcome remains unsatisfactory, partly due to drug resistance. Epigenetic regulation is closely associated with the conventional mechanism of MTAs in HCC treatment, the mechanism of which is also intimately associated with the TME, involving the vascular system, transport processes, and immune system. Evidence for resistance mechanisms in HCC has steadily been accumulating mainly with regard to sorafenib, regorafenib, and lenvatinib. However, evidence is still limited with respect to novel agents such as atezo + bev combination drugs, cabozantinib, and ramucirumab. Furthermore, the efficacy of new ICI-based therapy, such as tremelimumab plus durvalumab combination therapy, has been demonstrated and may be a promising therapeutic strategy to enhance the efficacy of MTAs or overcome drug resistance through changes in the tumor-immune microenvironment. In conclusion, it is imperative to elucidate the underlying mechanisms of drug resistance in HCC in order to provide new insights into future therapeutic strategies, such as the selection of combination therapy to overcome drug resistance and the selection of drugs to be used for subsequent therapy. We hope that such insights will ultimately contribute to improved outcomes in the multidisciplinary treatment of advanced HCC.

## Figures and Tables

**Figure 1 ijms-24-02805-f001:**
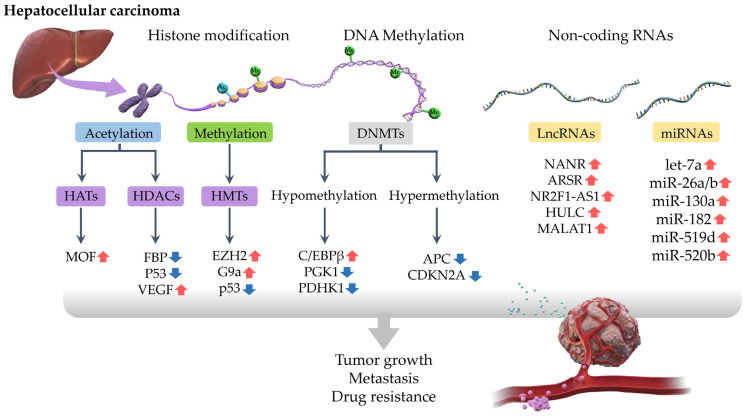
Schematic representation of epigenetic modifications in hepatocellular carcinoma (HCC). Histone proteins undergo various post-translational modifications by enzymes, such as histone acetyltransferases (HATs), histone deacetylases (HDACs), and histone methyltransferases (HMTs) to alter chromatin structure and modulate gene expression by providing transcriptional activators or repressors access to DNA sequences. Methylation of CpG sites, which are upstream regulatory elements of DNA found in promoters, enhancers, and transcription factor-binding sites, are mediated by enzymes such as DNA (cytosine-5)-methyltransferase 1 (DNMT1), DNMT3A, and DNMT3B, which suppress the activity and expression of tumor suppressor genes. Among non-coding RNA (ncRNA) that do not encode proteins or peptides, long ncRNAs (lncRNAs) and microRNAs (miRNAs) regulate gene expression at various levels, including transcription, translation, and protein function. These epigenetic modifications are also associated with various processes involved in tumor growth, metastasis, and drug resistance of HCC. Red arrows indicate increased expression and blue arrows indicate decreased expression. MOF, males absent on the first; FBP, fructose-1,6-bisphosphatase; VEGF, vascular endothelial growth factor; EZH2, enhancer zeste homolog 2; C/EBPβ, CCAAT/enhancer-binding protein-beta; PGK1, phosphoglycerate kinase 1; PDHK1, pyruvate dehydrogenase kinase 1; APC, adenomatosis polyposis coli; CDKN2A, cyclin-dependent kinase inhibitor 2A; NANR, HCC-associated lncRNA; ARSR, NR2F1-AS1, nuclear receptor subfamily 2 group F member 1-antisense RNA 1; HULC, highly upregulated in liver cancer; MALAT1, metastasis-associated lung adenocarcinoma transcript 1.

**Table 1 ijms-24-02805-t001:** Epigenetic modification and sorafenib resistance in hepatocellular carcinoma (HCC).

Molecules	Expression	Major Effects	References
DNA methylation			
MORC	Upregulated	MORC forms a complex with DNMT3 on the promoters of NF2 and KIBRA to cause DNA hypermethylation.	[164]
PRMT6	-	PRMT6 methylates CRAF and inhibits FRAS/RAF binding ability, thereby altering ERK-mediated nuclear transport of PKM2 and reducing sorafenib resistance.	[166]
BNIP3	Downregulated	Epigenetic silencing of BNIP3 is associated with sorafenib resistance. Promoter demethylation and restoration of BNIP3 can overcome sorafenib resistance.	[167]
Long non-coding RNAs			
H19	Downregulated	The promoter methylation of the H19 gene is associated with sorafenib resistance. Overexpression of H19 sensitizes HCC cells to sorafenib.	[165]
SNHG1	Upregulated	Sorafenib induces miR-21 to enter the nucleus and promote SNHG1 expression, leading to activation of AKT pathway and contributing to sorafenib resistance in HCC cells.	[168]
SNHG3	Upregulated	SNHG3 causes EMT of HCC cells through miR-128/CD151 cascade activation and is involved in sorafenib resistance.	[169]
SNHG16	Upregulated	Knockdown of SNHG16, which is upregulated in sorafenib-resistant HCC, improves sensitivity to sorafenib and functions as endogenous sponge for miR-140-5p.	[170]
NEAT1	Upregulated	NEAT1 negatively regulates miR-335 expression and inhibits the cMet-AKT pathway, which is associated with sorafenib resistance in HCC.	[171]
FOXD2-AS1	Downregulated	FOXD2-AS1 functions as a sponge for miR-150-5p and contributes to sorafenib resistance in HCC by suppressing TMEM9.	[172]
MicroRNAs			
miR-19a-3p	Upregulated	Aberrant expression of miR-19a-3p induces sorafenib resistance in HCC cells by regulating the PTEN/AKT pathway.	[86]
miR-181a	Upregulated	miR-181a induces sorafenib resistance in HCC via the suppression of RASSF1.	[173]
miR-221	Upregulated	mir-221 exerts anti-apoptotic activity by targeting caspase-3 and is involved in sorafenib resistance in HCC.	[174]
miR-374b	Downregulated	In sorafenib-resistant HCC cells and xenograft mice, miR-375b overcomes drug resistance by suppressing the hnRNPA1/PKM2 axis.	[175]
miR-494	Upregulated	Overexpression of miR-494 enhances sorafenib resistance in HCC cells by activating mTOR.	[176]
miR-622	Downregulated	MiR-622 contributes to the abrogation of drug resistance in sorafenib-resistant HCC cells by targeting KRAS and inhibiting RAF/ERK and PI3K/AKT signaling.	[177]

MORC, microrchidia 2; DNMT, DNA (cytosine-5)-methyltransferase; NF2, neurofibromatosis type 2; KIBRA, kidney and brain protein; PRMT6, N-methyltransferase 6; FRAS, Fraser extracellular matrix complex subunit; ERK, extracellular signal-regulated kinase; PKM2, pyruvate kinase M2; Bcl-2 interacting protein 3; SNHG, small nucleolar RNA host gene 1; EMT, epithelial–mesenchymal transition; NEAT1, nuclear-enriched abundant transcript 1; FOXD2-AS1, forkhead box protein D2-antisense RNA 1; PTEN, phosphatase and tensin homolog; hnRNPA1, heterogeneous nuclear ribonucleoprotein A1; mTOR, mammalian target of rapamycin; KRAS, v-Ki-ras2 Kirsten rat sarcoma viral oncogene homolog; PI3K, phosphatidylinositol 3-kinase.

**Table 2 ijms-24-02805-t002:** Tumor microenvironment and regorafenib resistance in hepatocellular carcinoma (HCC).

Molecules	Expression	Major Effects	Reference
Epithelial–mesenchymal transition			
Pin1	Upregulated	Pin1 regulates the expression of EMT-related molecules such as E-cadherin and promotes HCC progression, invasion, and metastasis in regorafenib resistance of HCC.	[197]
Cell cycle			
FOXM1	Upregulated	FOXM1 is overexpressed in regorafenib-resistant HCC cells and elevated FOXM1 expression, correlating with drug resistance and decreased survival.	[199]
Apoptosis			
TOP2A	Upregulated	Elevated TOP2A expression correlates with regorafenib resistance and poor prognosis in patients with HCC.	[200]
Others			
TGF-β	Upregulated	Regorafenib-resistant HCC cells deactivate Wnt/β-catenin signaling and activate TGF-β signaling. Regorafenib resistance is restored by TGF-β type 1 receptor inhibition.	[148]
HK1	Upregulated	Regorafenib-resistant HCC cells increase the expression of HK1, which catalyzes glucose metabolism, and HK1 expression correlates with drug resistance.	[201]
ATF3	Upregulated	ATF3-mediated upregulation of IL-6α induces multifunctional cytokines and promotes regorafenib resistance against HCC cells.	[202]
SphK2	Upregulated	Regorafenib-resistant HCC cells show high expression of SphK2. SphK2/S1P causes regorafenib resistance via the activation of NF-κB and STAT3.	[203]

Pin1, peptidyl-prolyl cis-trans isomerase 1; EMT, epithelial–mesenchymal transition; FOXM1, forkhead box protein M1; TOP2A, topoisomerase II alpha; TGF-β, transforming growth factor beta; HK1, hexokinase 1; ATF3, activating transcription factor 3; IL-6α, interleukin-6 alpha; SphK2, sphingosine kinase 2; S1P, sphingosine-1-phosphate; NF-κB, nuclear factor-kappa B; STAT3, signal transducer and activator of transcription 3.

**Table 3 ijms-24-02805-t003:** Tumor microenvironment and lenvatinib resistance in hepatocellular carcinoma (HCC).

Molecules	Expression	Major Effects	Reference
Non-coding RNAs			
lncXIST	Upregulated	XIST promotes lenvatinib resistance in HCC cells via the activation of EZH2/NOD2/ERK axis.	[211]
lncAC026401.3	Upregulated	AC026401.3 is upregulated in HCC tissues and correlates with poor prognosis in HCC patients. AC026401.3 interacts with OCT1 to activate E2F2 and enhances lenvatinib resistance in HCC.	[212]
miR-128-3p	Downregulated	Downregulation of miR-128-3p is involved in lenvatinib resistance via AKT and ERK.	[213]
circPAK1	Upregulated	CircRAK1 is highly expressed in HCC cells and tissues and correlates with poor prognosis in HCC patients. CircPAK1 is transported by exosomes to induce lenvatinib resistance.	[214]
circMED27	Upregulated	Serum circMED27 is significantly elevated in HCC patients, correlating with poor prognosis. Competing with miR655-3p, cirMED27 upregulates USP28 to promote lenvatinib resistance.	[215]
Transport processes			
BCRP	Upregulated	BCRP transporter expression is elevated in lenvatinib-resistant HCC cells. Activation of EGFR-, MEK/ERK-, and PI3K/Akt-signaling pathways are associated with lenvatinib resistance.	[216]
ABCB1	Upregulated	Activation of EGFR- and STAT3/ABCB1-signaling pathways and enhancement of exocytosis cause lenvatinib resistance in HCC cells.	[217]
Hypoxia			
NRP1	-	NRP1 gene silencing significantly enhances the anti-tumor effect of lenvatinib resistance. HIF1α directly regulates NRP1 expression in hypoxic microenvironment.	[218]
HIF-1α	Upregulated	Under hypoxic conditions, transcription factors, including HIF-1α, are induced, increasing fibronectin expression, leading to lenvatinib resistance in HCC.	[219]
Others			
ERK1	Upregulated	Activation of ERK1 signaling is observed in lenvatinib-resistant HCC cells. Cisplatin shows an effective anti-tumor effect on lenvatinib-resistant HCC cells via ATM/ATR-Chk1/Chk2 signaling.	[220]
ERK	Upregulated	Activation of MAPK/MEK/ERK signaling and increased expression of EMT markers are observed in lenvatinib-resistant HCC cells.	[221]
ERK	Upregulated	VEGFR2 expression and its downstream RAS/MEK/ERK signaling are elevated in renvatinib-resistant HCC cells.	[222]
ERK	Upregulated	Knockout of DUSP4, a gene associated with lenvatinib resistance, activates ERK/MEK signaling at the phosphorylation level and induces lenvatinib resistance in HCC cells.	[223]

XIST, X-inactive specific transcript; EZH2, enhancer of zeste homolog 2; NOD2, nucleotide-binding oligomerization domain containing 2; ERK, extracellular signal-regulated kinase; OCT1, organic cation uptake transporter 1; circPAK1, circulating p2 of 1-activated kinase 1; circMED27, circulating RNA mediator complex subunit 27; USP28, ubiquitin-specific peptidase 28; BCRP, breast cancer resistance protein; EGFR, epidermal growth factor receptor; MEK, mitogen-activated protein kinase; PI3K, phosphatidylinositol-3 kinase; ABCB1, ATP-binding cassette subfamily B member 1; STAT3, signal transducer and activator of transcription 3; NRP1, neuropilin-1; HIF-1α, hypoxia-inducible factor-1α; ATM, ataxia telangiectasia mutated; ATR, ataxia telangiectasia and Rad3-related protein; Chk1, checkpoint kinase 1; Chk2, checkpoint kinase 2; MAPK, mitogen-activated protein kinase; VEGFR2, vascular endothelial growth factor receptor 2; DUSP4, dual specificity phosphatase 4.

## Data Availability

Not applicable.

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
