# Peer review of "The Roles of Epigenetic Regulation and the Tumor Microenvironment in the Mechanism of Resistance to Systemic Therapy in Hepatocellular Carcinoma"

_ijms, 2023, doi:10.3390/ijms24032805_

Round 1

Reviewer 1 Report

To Authors:

The manuscript ijms-2150582-peer-review-v1 “The roles of epigenetic regulation and the tumor microenvironment in the mechanism of resistance …… by Masaki et all is very comprehensive and the reviewed and cited most of the relevant HCC disease discipline.

This reviewer does not have any major concerns that need major revision but the following minor suggestions to fulfill in order to consider for a publication in IJ Molecular sciences as follows:

Abstract:

Line 6, sentence to read as; Systemic therapies are recommended for unresectable HCC (uHCC) with major … 

Section 3.3 “Immune System” paragraph 2, include this sentence after reference 118: remelimumab (anti-cytotoxic T lymphocyte–associated antigen 4) plus durvalumab (anti–programmed cell death ligand-1) demonstrated promising clinical activity and safety in a Phase 2 trial of uHCC.

Conclusion section:

Add we lines about uHCC, current Phase 2, 3 trials status of these immunotherapy agents: remelimumab and durvalumab and therapy outcome minimizing drug resistance over small molecules such as Sorafenib, Regorafenib, and Lenvatinib where authors heavily stressed on these drugs limiting on immunotherapeutic agents to justify the title which is on TME

References:

Include this reference:
NEJM Evid 1 (8) 2022: Tremelimumab plus Durvalumab in Unresectable Hepatocellular Carcinoma

Author Response

Response to reviewer 1 comments

(1) Abstract: Line 6, sentence to read as; Systemic therapies are recommended for unresectable HCC (uHCC) with major … 

Response: Thank you very much for your comments.

According to your comment, we have altered the corresponding part in the abstract.

(2) Section 3.3 “Immune System” paragraph 2, include this sentence after reference 118: remelimumab (anti-cytotoxic T lymphocyte–associated antigen 4) plus durvalumab (anti–programmed cell death ligand-1) demonstrated promising clinical activity and safety in a Phase 2 trial of uHCC.

Response: Thank you very much for your comments.

As you have indicated, tremelimumab plus durvalumab is a promising combination immunotherapy with anti-cytotoxic T lymphocyte-associated antigen-4 (CTLA-4) and anti-programmed cell death ligand-1 (PD-L1) antibodies in unresectable hepatocellular carcinoma (HCC). The combination of tremelimumab and durvalumab demonstrated clinical activity and safety in a phase II trial of unresectable HCC (Kelley RK et al. J Clin Oncol. 39(27): 2991-3001, 2021). Furthermore, in a phase III trial, single tremelimumab regular interval durvalumab regimen demonstrated superior overall survival (OS) to sorafenib in patients with unresectable HCC and no previous systemic treatment. Durvalumab monotherapy was also shown non-inferiority to sorafenib (Abou-Alfa GM et al. NEJM Evid. 1(8), 2022).

According to your comment, we have altered the corresponding part in section 3.

(3) Conclusion section: Add we lines about uHCC, current Phase 2, 3 trials status of these immunotherapy agents: remelimumab and durvalumab and therapy outcome minimizing drug resistance over small molecules such as Sorafenib, Regorafenib, and Lenvatinib where authors heavily stressed on these drugs limiting on immunotherapeutic agents to justify the title which is on TME

 Response: Thank you very much for your comments.

As you have pointed out, recent studies have shown that long-term treatment with the combination of immune checkpoint inhibitor (ICI) and growth factor inhabitation, such as atezolizumab plus bevacizumab combination, is superior to conventional tyrosine kinase inhibitor (TKI) monotherapy in terms of OS and life-year gain (Finn R et al. N Engl J Med. 382(20): 1894-1905, 2020).

In ICI-based therapy, the combination of tremelimumab plus durvalumab was recently shown to significantly improve median OS compared to sorafenib in the phase III trials (Abou-Alfa, GM et al. NEJM Evid 1(8), 2022). Further, several clinical trials have been conducted on ICI-based therapies in later-line treatment, and while some promising results have been shown in nivolumab monotherapy (Yau T et al. Lancet Oncol. 23: 77-90, 2022), nivolumab plus ipilimumab (Yau T et al. JAMA Oncol. 6: e204564, 2020), and pembrolizumab (Qin S et al. Clin Oncol. 40: 383, 2022), the evidence for clinical use appears to be insufficient. ICI-based therapies ma be a promising therapeutic strategy to enhance the therapeutic efficacy of TKIs such as sorafenib, regorafenib, and lenvatinib, or to overcome drug resistance through changes in the tumor immune microenvironment.

According to your comment, we have altered the corresponding part of the conclusion.

Reviewer 2 Report

In this paper, Kyoko etal. Summarized “The roles of epigenetic regulation and the tumor microenvironment in the mechanism of resistance to systemic therapy in hepatocellular carcinoma”. As we know the liver cancer is the 6th common tumor and it is a kind of easily cause death cancer worldwide. I have some questions need to concern. This is a well written review for introduce the epigenetic changed in liver cancer. The details as follows:

Major concern,

Epigenic regulation many biological processes, it should be also playing a key role in the liver cancer, but how does it change not so clear. Recently, many scientists found the mainly three parts in epigenetics, including DNA structure changed by DNA methylation, histone acetylation, and lncRNA regulation of cancer DNA process. 5mC is a major DNA modification by methylation of CpG island in the genomic DNA. This modification can repress the DNA replication and this process was finished by DNMT1 enzyme. In liver cancer, NA hypomethylation caused the genomic instability, frequent mutations, and transitions occurring at inactive sites in chromatin regions, often in specific CpG islands, repeated DNA sequences, and intergenic regions. Many genes will be modification by DNA methylation, which genes play the key role for liver cancer metastasis?

Histone modification also another epigenetics that happened in histone proteins. Does H3 H4 histone acetylation also changed in liver cancer?

Many studies have showed the lncRNA have positive and negative effect on liver tumor? For liver cancer, which one can be used as a target to treatment of liver cancer?

Tumor microenvironment is important for tumor growth and metastasis. How does treatment of tumor by targets the liver microenvironment should be give more discussion.

For the immune cells in the tumor microenvironment, how does they promote or destroy the tumor growth should give some more discussion.

Try to more focus on the epigenetics changed relative to liver cancer, the whole review article list many other cancer, and looks unnecessary to introduce more other cancer, because a large number of papers introduced difference type of cancer treatment.

Minor edit,

Line 33, “affected by” should use “diagnosed as” .

Line 59, “regimens” need to reedit as “chemicals or drugs”

Please find a native English speaker to edit the full paper, and make it more readable.

Author Response

Response to reviewer 2 comments

(1) Epigamic regulation many biological processes, it should be also playing a key role in the liver cancer, but how does it change not so clear. Recently, many scientists found the mainly three parts in epigenetics, including DNA structure changed by DNA methylation, histone acetylation, and lncRNA regulation of cancer DNA process. 5mC is a major DNA modification by methylation of CpG island in the genomic DNA. This modification can repress the DNA replication and this process was finished by DNMT1 enzyme. In liver cancer, NA hypomethylation caused the genomic instability, frequent mutations, and transitions occurring at inactive sites in chromatin regions, often in specific CpG islands, repeated DNA sequences, and intergenic regions. Many genes will be modification by DNA methylation, which genes play the key role for liver cancer metastasis?

Response: Thank you very much for your comments.

Epigenetic regulation of transcription involves cross-talk between DNA methylation, chromatin remodeling enzymes, and histone modifications. Dysfunction in DNA methylation has been shown to be closely associated with carcinogenesis, autoimmune diseases, and fibrosis diseases (Bakusic J et al. J Psychosom Res. 92: 33-44, 2017). As you pointed out, aberrant methylation, especially of tumor suppressors, is characteristic of various cancers, including hepatocellular carcinoma (HCC), especially genomic instability (Villanueva A at al. Hepatology. 61(6): 1945-1956, 2015), frequent mutations, metastases occurring at inactive sites in chromatin regions (Hama N et al. Nat Commun. 9(1): 1643, 2018), often specific CpG islands, repeated DNA sequences and intergenic regions are responsible for DNA hypomethylation in HCC.

There are several reports on DNA methyltransferase (DNMT)-mediated epigenetic changes with regulation of HCC metastasis. In terms of the role of DNMT1, knockdown of non-collagenous bone matrix protein osteopontin (OPN) in CD133+/CD44+ HCC cells, which have cancer stem cell properties, suppresses DMNT1 expression and promotes HCC metastasis (Gao X et al. J. Exp. Clin. Cancer Res. 37: 179, 2018). Epigenetic upregulation of c-Met, the receptor for hepatocyte growth factor (HGF) is associated with HCC progression and metastasis in the TME, and it was shown that a significant decrease in DNA methylation during hematogenous metastasis of HCC correlated with increased c-MET expression in circulating tumor cells (Ogunwobi OO et al. Clin. Cancer Res. Off. J. Am. Assoc. Cancer Res. 19: 2310-2318, 2013). Other reports showed that induction of DNMT1 expression by HGF leaded to DNA hypermethylation of tumor suppressor genes such as myocardin, pannexin 2, and LIN homeobox 9 genes, which was associated with HCC metastasis (Xie CR et al. Mol. Med. Rep. 12: 4250-4258, 2015). DNMT3 has also been shown to promote HCC metastasis and invasion by epigenetic regulation of the metastasis-associated protein 1 (MTA1) gene, and in hepatitis B virus (HBV)-associated HCC, HBV X mobilizes DNMT3a and DNMT3b to increase promoter methylation and enhance MTA1 expression (Lee MH et al. Oncogenesis. 1: e25, 2012). These studies provide important evidence that DNA methylation catalyzed by DNMTs is relevant to the regulation and molecular mechanisms of the HCC transition.

As you suggested, we have altered the relevant part of the section to make the context clear.

(2) Histone modification also another epigenetics that happened in histone proteins. Does H3 H4 histone acetylation also changed in liver cancer?

Response: Thank you very much for your comments.

In histone modifications in HCC, histone H3 methylation and acetylation have been most intensively studied. H3K4 trimethylation and H3 acetylation are abnormally high and H3K27 trimethylation was low in promoters in vasohibin 2 (VASH2) gene, which functions as a growth factor in HCC (Li D et al. Biosci Rep. 39(8), 2019). Interestingly, suppression of VASH2 expression inhibited HCC proliferation and induces apoptosis. In another report on histone H3 acetylation, it was shown that insulin induced major transcriptional factors such as sterol regulatory element-binding transcription factor 1c (SREBP-1c) and carbohydrate responsive-element binding protein (ChREBP) binding with sterol regulatory elements (SRE) or carbohydrate responsive-elements (ChORE) of the fatty acids synthase (FASN) promoter and induces FASN expression in normal tissue, while hyperacetylation of histone H3 and H4 impaired SREBP-1c-SRE and ChREBP-ChORE binding on FASN promoter and HCC becomes insulin resistant (Du X et al. Biochem. Biophys. Res. Commun. 483: 409-417, 2017). In vitro experiment has shown that HCC cells have relatively lower nucleosome density with histone H3K9 acetylation than controls, regardless of transcriptional activation status, which may play an important role in initiating HCC development (Nishida H et al. Chromosome Res. 14: 203-211, 2006). On the other hand, for Histone 3 lysine methylation, it has been shown that the higher the level of H3K4 trimethylation in HCC, the worse the prognosis for HCC (He C et al. Hum. Pathol. 43: 1425-1435, 2012). The histone methyltransferase mixed-lineage leukemia (MLL) causes H3K4 trimethylation and the MLL-E-twenty-26 transcription factor 2 complex occupies the matrix metallopeptidase 1 (MMP1)/MMP3 gene promoter, resulting in activation of MMP1/MMP3 expression. This means that MLL-mediated H3K4 trimethylation is required for HCC proliferation and metastasis by HGF (Takeda S et al. J. Clin. Invest. 123: 3154-3165, 2013). In another report, HBV X protein was shown to promote hepatocarcinogenesis in a cellular model by inducing H3K9me3 at the p16 promoter via down-regulation of the demethylase jumonji domain containing 2B genes, which promotes repression of p16 gene expression (Wang D et al. Exp. Mol. Pathol. 99: 399-408, 2015).

These finding is important and have been added to the appropriate section.

(3) Many studies have showed the lncRNA have positive and negative effect on liver tumor? For liver cancer, which one can be used as a target to treatment of liver cancer?

Response: Thank you very much for your comments.

Abnormal expression of several lncRNAs has been observed in HCC, and these lncRNAs have been reported to interact with DNA, RNA, and proteins to form complexes that regulate the expression of target genes. Lung cancer-associated transcript 1 (LUCAT1) is a lncRNA known to regulate growth and metastasis in various cancer types. LUCAT1 expression is also enhanced in tissues and cells of HCC, and loss- and gain-of-function studies have shown that LUCAT1 promotes growth and metastasis in HCC (Lou Y et al. J Cell Mol Med. 23(3): 1873-1884, 2019). lncRNA MCM3AP antisense RNA 1 (MCM3AP-AS1) was also overexpressed in HCC tissues and cell lines and positively correlated with large tumor size, high tumor grade, advanced tumor stage, and poor prognosis, indicating that knockdown of MCM3AP-AS1 inhibits HCC growth. Furthermore, MCM3AP-AS1 directly binds to miR-194-5p and promotes the expression of its target gene forkhead box A1 (FOXA1), which was the anti-tumor mechanism of HCC (Wang Y et al. Mol Cancer. 18(1): 28, 2019). Thus, quite a few lncRNAs involved in HCC progression have been identified, and the use of several antitumor compounds that target these lncRNAs may be the best therapeutic strategy. Overexpression of lncRNA metastasis associated in lung adenocarcinoma transcript 1 (MALAT1) was observed in HCC, and while that the lncRNA promoted HCC proliferation via overexpression of surtuin 1 gene (Hou Z et al. tumor biology. 39, 2017), it was also shown that gallic acid downregulate MALAT1, resulting in Wnt/β-catenin signal inhibition and suppressing HCC progression (Shi CJ et al. Front. Pharmacol. 12, 2021). In addition, there are several reports on the antitumor effects of targeting lncRNAs with melatonin, which is used in the treatment of HCC. Melatonin was shown to increase lncRNA RAD51 antisense RNA 1 (RAD51-AS1) expression levels, mediate drug sensitivity, and inhibit HCC progression (Chen CC et al. Cancers. 10: 320, 2018). Another study showed that melatonin promotes FOXA2 expression levels and upregulates lncRNA Carbamoyl-phosphate synthetase 1 (CPS1) in the downregulation of hypoxia-induced factor-1α, inhibiting epithelial-mesenchymal transition and HCC carcinogenesis (Wang TH et al. Oncotarget. 8: 82280-82293, 2017), which may be a therapeutic strategy to target IncRNA.

As you suggested, we have altered the relevant part of the section to make the context clear.

(4) Tumor microenvironment is important for tumor growth and metastasis. How does treatment of tumor by targets the liver microenvironment should be give more discussion.

For the immune cells in the tumor microenvironment, how does they promote or destroy the tumor growth should give some more discussion.

Try to more focus on the epigenetics changed relative to liver cancer, the whole review article list many other cancer, and looks unnecessary to introduce more other cancer, because a large number of papers introduced difference type of cancer treatment.

Response: Thank you very much for your comments.

With respect to the vascular system in TME, several angiogenesis stimulating factors such as vascular endothelial growth factor (VEGF), fibroblast growth factor (FGF), platelet-derived growth factor (PDGF), their receptors, and endoglin are associated with HCC growth and are therapeutic targets for HCC. Among these molecules, VEGF strongly promotes angiogenesis, and in fact, most of the MTAs approved to date for advanced HCC, such as sorafenib, regorafenib, and lenvatinib target the VEGF/VEGF receptor (VEGFR) angiogenic pathway. Circulating VEGF levels were shown to be elevated in HCC and correlated with tumor angiogenesis and progression, and an association between high tumor microvessel density and increased local and circulating VEGF with rapid disease progression and poor prognosis (Poon RT et al. Ann Surg. 238: 9-28, 2003), supporting efficacy of targeting the VEGF pathway in HCC therapy. In a report on FGF and VEGF crosstalk, FGF-2 and VEGF-A have been associated with increased capillary sinusoids in HCC tumor angiogenesis (Motoo Y et al. Oncology. 50: 270-274, 1993), and FGF stimulation modulates expression of integrins that regulate endothelial cells in the MTA and alter cell parameters required for angiogenesis. Placental growth factor (PLGF), which belongs to the VEGF family, can also be therapeutic targets for HCC (Xu HX et al. Int J Cancer. 128(7): 1559-1569, 2011).

The immune microenvironment of HCC is riche in inflammatory chemokines, cytokines, and immunosuppressive molecules that create a strongly immunosuppressive TME and an important role in reorganizing TME, mediating intercellular cross-talk, and promoting immune evasion. The most studied immune checkpoints in HCC are cytotoxic T lymphocyte protein 4 (CTLA-4), programmed cell death protein-1 and its ligands (PD-1, PD-L1), and mucin domain containing molecule 3 (Tim-3). CTLA4 is an inhibitory co-receptor constitutively present on Tregs and plays an important role in regulating CD4+ T cell function, and in HCC, as in other cancer types, it inhibits T cell proliferation through the recognition and differentiation of tumor-associated antigens (Kudo M. Oncology. 92(Suppl 1): 50-62, 2017). Furthermore, in HCC tissues, CTLA-4 contributes to tumor growth by promoting immunosuppression through induction of Treg activity and production of indoleamine-2,3-dioxygenase and IL-10 in DCs (Han Y et al. Hepatology. 59(2): 567-579, 2014). PD-1 is a regulatory immunoglobulin expressed on activated CD4+ and CD8+ T cells, B cells, and NK cells and plays an important role in maintaining immune tolerance and suppressing T lymphocyte cytotoxicity (Wei SC et al. Cell. 170(6): 1120-1133, 2017)). It is also known that upregulation of PD-L1 on HCC cells induced by various cytokines, especially IFN-γ, contributes to impaired anti-tumor immunity and promotes apoptosis of CD8+ T cells (Shi F et al. Int J Cancer. 128(4): 887-896, 2011). Tim-3 is a transmembrane immunoglobulin expressed on IFN-γ-secreting Th1 cells, NK cells Tim-3 expression is increased in T cells infiltrating chronic HBV infection, and the Tim-3/galectin-9 pathway is associated with poor prognosis in patients with HBV-associated HCC (Mengshol JA et al. PLos One. 5(3): e9504, 2010). The clinical value of these immune checkpoint molecules in HCC needs to be further elucidated.

As you suggested, these finding is important and have been added to the appropriate section.

(5) Minor edit,

Line 33, “affected by” should use “diagnosed as” .

Line 59, “regimens” need to reedit as “chemicals or drugs”

Response: Thank you very much for your comments.

As you indicated, we have altered the relevant part of the text.

(6) Please find a native English speaker to edit the full paper, and make it more readable

Response: Thank you very much for your comments.

As you indicated, we worked with native speakers to check the entire manuscript for errors, and again asked the proofreading company to revise the manuscript to improve its expression.

We believe this revision will make our manuscript more readable.
